# Mutations in L-type amino acid transporter-2 support *SLC7A8* as a novel gene involved in age-related hearing loss

Meritxell Espino Guarch[1,2,3†]*, Mariona Font-Llitjós[2,4†], Silvia Murillo-Cuesta[4,5,6], Ekaitz Errasti- Murugarren[3,4], Adelaida M Celaya[4,5], Giorgia Girotto[7,8], Dragana Vuckovic[7,8], Massimo Mezzavilla[1], Clara Vilches[2], Susanna Bodoy[3,4], Ignasi Sahún[4,9], Laura González[2,4], Esther Prat[2,4,10], Antonio Zorzano[3,11,12], Mara Dierssen[4,9], Isabel Varela-Nieto[4,5,6‡], Paolo Gasparini[7,8‡], Manuel Palacín[3,4,11‡]*, Virginia Nunes[2,4,10‡]*

[1]Experimental Genetics, Sidra Medical and Research Center, Doha, Qatar; [2]Genes, Disease and Therapy Program, Molecular Genetics Laboratory - IDIBELL, Barcelona, Spain; [3]Institute for Research in Biomedicine (IRB Barcelona), The Barcelona Institute of Science and Technology, Barcelona, Spain; [4]Biomedical Research Networking Centre on Rare Diseases (CIBERER), Institute of Health Carlos III, Barcelona, Spain; [5]Alberto Sols Biomedical Research Institute (CSIC/UAM), Madrid, Spain; [6]Hospital La Paz Institute for Health Research (IdiPAZ), Madrid, Spain; [7]Department of Medicine, Surgery and Health Sciences, University of Trieste, Trieste, Italy; [8]Medical Genetics, Institute for Maternal and Child Health – IRCCS "Burlo Garofolo", Trieste, Italy; [9]Center for Genomic Regulation (CRG), The Barcelona Institute of Science and Technology, Barcelona, Spain; [10]Genetics Section, Physiological Sciences Department, Health Sciences and Medicine Faculty, University of Barcelona, Barcelona, Spain; [11]Biochemistry and Molecular Biomedicine Department, Faculty of Biology, University of Barcelona, Barcelona, Spain; [12]Biomedical Research Networking Centre on Diabetes and Associated Metabolic Diseases (CIBERDEM), Barcelona, Spain

*For correspondence:
mespinoguarch@sidra.org (MEG);
manuel.palacin@irbbarcelona.org (MPi);
vnunes@idibell.cat (VN)

†These authors contributed equally to this work
‡These authors also contributed equally to this work

Competing interests: The authors declare that no competing interests exist.

**Abstract** Age-related hearing loss (ARHL) is the most common sensory deficit in the elderly. The disease has a multifactorial etiology with both environmental and genetic factors involved being largely unknown. SLC7A8/SLC3A2 heterodimer is a neutral amino acid exchanger. Here, we demonstrated that SLC7A8 is expressed in the mouse inner ear and that its ablation resulted in ARHL, due to the damage of different cochlear structures. These findings make SLC7A8 transporter a strong candidate for ARHL in humans. Thus, a screening of a cohort of ARHL patients and controls was carried out revealing several variants in *SLC7A8*, whose role was further investigated by in vitro functional studies. Significant decreases in SLC7A8 transport activity was detected for patient's variants (p.Val302Ile, p.Arg418His, p.Thr402Met and p.Val460Glu) further supporting a causative role for SLC7A8 in ARHL. Moreover, our preliminary data suggest that a relevant proportion of ARHL cases could be explained by SLC7A8 mutations.
DOI: https://doi.org/10.7554/eLife.31511.001

## Introduction

Age-related hearing loss (ARHL) or presbycusis is one of the most prevalent chronic medical conditions associated with aging. Indeed, more than 30% of people aged over 65 years suffer ARHL

**eLife digest** Age-related hearing loss affects about one in three individuals between the ages of 65 and 74. The first symptom is difficulty hearing high-pitched sounds like children's voices. The disease starts gradually and worsens over time. Changes in the ear, the nerve that connects it to the brain, or the brain itself can cause hearing loss. Sometimes all three play a role. Genetics, exposure to noise, disease, and aging may all contribute. The condition is so complex it is difficult for scientists to pinpoint a primary suspect or develop treatments.

Now, Guarch, Font-Llitjoś et al. show that errors in a protein called SLC7A8 cause age-related hearing loss in mice and humans. The SLC7A8 protein acts like a door that allows amino acids – the building blocks of proteins – to enter or leave a cell. This door is blocked in mice lacking SLC7A8 and damage occurs in the part of their inner ear responsible for hearing. As a result, the animals lose their hearing. Next, Guarch, Font-Llitjoś et al. scanned the genomes of 147 people from isolated villages in Italy for mutations in the gene for SLC7A8. The people also underwent hearing tests.

Mutations in the gene for SLC7A8 that partially block the door and prevent the flow of amino acids were found in people with hearing loss. Some mutations in SLC7A8 that allow the door to stay open where found in people who could hear. The experiments suggest that certain mutations in the gene for SLC7A8 are likely an inherited cause of age-related hearing loss. It is possible that other proteins that control the flow of amino acids into or out of cells also may play a role in hearing. More studies are needed to see if it is possible to fix errors in the SLC7A8 protein to delay or restore the hearing loss.

DOI: https://doi.org/10.7554/eLife.31511.002

(*Gates and Mills, 2005*; *Huang and Tang, 2010*; *Van Eyken et al., 2007*). Clinically, ARHL is defined as a progressive bilateral sensorineural impairment of hearing in high sound frequencies mainly caused by a mixture of 3 pathological changes: loss of the hair cells of the organ of Corti (sensory), atrophy of the stria vascularis (metabolic) and degeneration of spiral ganglion neurons (SGN), as well as the central auditory pathway (neural) (*Gates and Mills, 2005*; *Schuknecht, 1955*; *Yamasoba et al., 2013*). ARHL has a complex multifactorial etiology with both genetic and environmental factors contributing (*Cruickshanks et al., 2010*; *Christensen et al., 2001*). Although most people lose hearing acuity with age, it has been demonstrated that genetic heritability affects the susceptibility, onset and severity of ARHL (*Wingfield et al., 2007*; *Cruickshanks et al., 2001*; *Gates et al., 1999*; *Karlsson et al., 1997*; *Cruickshanks et al., 1998*). Unfortunately, the complexity of the pathology coupled with highly variable nature of the environmental factors, which cause cumulative effects, increases the difficulty in identifying the genetic contributors underlying ARHL. Most of the findings from genome-wide association studies (GWAS) performed into adult hearing function could neither be replicated between populations, nor the functional validation of those candidates be confirmed (*Dawes and Payton, 2016*). Mouse models, including inbred strains, have been essential for the identification of several defined loci that contribute to ARHL (*Bowl and Dawson, 2015*).

SLC7A8/SLC3A2 is a Na$^+$-independent transporter of neutral amino acids that corresponds to system L also known as LAT2 (**L**-type **A**mino acid **T**ransporter-**2**) (*Pineda et al., 1999*; *Rossier et al., 1999*; *Oxender and Christensen, 1963*). SLC7A8 is the catalytic subunit of the heterodimer and mediates obligatory exchange with 1:1 stoichiometry of all neutral amino acids, including the small ones (e.g. alanine, glycine, cysteine and serine), which are poor substrates for SLC7A5 (18), another exchanger with system L activity. Functional data indicate that the role of SLC7A8 is to equilibrate the relative concentrations of different amino acids across the plasma membrane instead of mediating their net uptake (*Pineda et al., 1999*; *Meier et al., 2002*; *Verrey, 2003*). The SLC7A8/SLC3A2 heterodimer is primarily expressed in renal proximal tubule, small intestine, blood-brain barrier and placenta, where it is thought to have a role in the flux of amino acids across cell barriers (*Rossier et al., 1999*; *Bauch et al., 2003*; *Kanai and Endou, 2001*; *del Amo et al., 2008*). So far, SLC7A8 research has been focused mainly on amino acid renal reabsorption. However, in vitro studies demonstrated that SLC7A8 could have a role in cystine efflux in epithelial cells and the in vivo

deletion of *Slc7a8* in a mouse model showed a moderate neutral aminoaciduria (*Braun et al., 2011*), suggesting compensation by other neutral amino acid transporters.

Therefore, in order to better understand the physiology of SLC7A8, we generated null *Slc7a8* knockout mice (*Slc7a8$^{-/-}$*) (*Font-LLitjós, 2009*) and (*Figure 1—figure supplement 1A*). Here, we describe the detection of a hypoacusic phenotype in the *Slc7a8$^{-/-}$* mouse model and demonstrate that novel loss-of-function SLC7A8 mutations constitute a primary cause in the development of ARHL in a cohort of elderly people from two isolated villages in Italy.

## Results

### *Slc7a8* ablation causes ARHL

SLC7A8 is highly expressed in the kidney, intestine and brain, and neither full-length nor truncated SLC7A8 protein were detected in membrane samples of *Slc7a8$^{-/-}$* mice (*Figure 1A*). The Allen Brain Atlas (*Allen Institute for Brain Science, 2004*) localizes mouse brain SLC7A8 to the cortical subplate, cerebellum, thalamus and olfactory bulb. Our results showed that SLC7A8 protein was localized to the plasma membrane of neuronal axons in different brain regions such as, the choroid plexus, subfornical organ, cerebral cortex and hypothalamus by immunohistochemistry (*Figure 1—figure supplement 2A*). This specific localization in the brain pointed to the possibility that the absence of the transporter could potentially lead to neurological disorders. Behavioral screening showed that absence of SLC7A8 in mice does not affect either learning or memory (*Figure 1—figure supplement 3*). In contrast, a significant reduction in latency was observed in the rotarod acceleration test indicating impairment in motor coordination in *Slc7a8$^{-/-}$* mice (*Figure 1—figure supplement 3G*). Reaffirming poorer motor coordination performance in the *Slc7a8$^{-/-}$* mice, an increased exposure to shock on the treadmill was also observed (*Figure 1—figure supplement 3B*). Interestingly, a marked impairment was observed in the pre-pulse inhibition of acoustic startle response, which assesses the response to a high intensity acoustic stimulus (pulse) and its inhibition by a weaker pre-pulse. The response to a 120 dB single-pulse was significantly reduced in *Slc7a8$^{-/-}$* mice (*Figure 1B*). The higher threshold required for responding to the acoustic stimulus in the PPI tests in *Slc7a8$^{-/-}$* animals could potentially be indicative of a hearing impairment or to a defect in the stress response signaling.

Response to stress is modulated by the hypothalamic-pituitary-adrenal axis via the release of corticosterone from the adrenal cortex (*Smith and Vale, 2006*). As SLC7A8 is expressed in the murine pituitary gland (*Figure 1—figure supplement 2A* and S3H), plasma corticosterone levels under stressing conditions were analyzed. No differences were observed in corticosterone levels at either basal conditions, nor under restraint stress in the *Slc7a8$^{-/-}$* group, indicating a normal stress response in the absence of SLC7A8 (*Figure 1—figure supplement 3I*). Thus, a hearing impairment in *Slc7a8$^{-/-}$* animals was considered the most probable cause of the differences observed in the acoustic startle response test (*Figure 1B*). The impact of the ablation of SLC7A8 on the auditory system was tested initially on mice with a mixed C57BL6/J-129Sv genetic background.

Auditory brainstem response (ABR) recording, which evaluates the functional integrity of the auditory system, was performed in *Slc7a8$^{-/-}$* mice. Reinforcing our hypothesis, adult 4- to 6-month-old *Slc7a8$^{-/-}$* mice showed significantly higher (p≤0.01) ABR thresholds in response to click stimulus, compared with age matched *Slc7a8$^{+/-}$* and wild type mice, which maintain normal hearing thresholds (*Figure 1C–E*). The hearing loss observed in *Slc7a8$^{-/-}$* mice affected the highest frequencies tested (20, 28 and 40 kHz) (*Figure 1F*). The analysis of latencies and amplitudes of the ABR waves in response to click stimuli, showed increased latency and decreased amplitude of wave I, but similar II-IV interpeak latency, in the *Slc7a8$^{-/-}$* mice when compared with the other genotypes, pointing to a hypoacusis of peripheral origin without affectation of the central auditory pathway (*Figure 1—figure supplement 4A to D*).

Mice were grouped according to genotype, age and ABR threshold level and descriptive statistics calculated, showing that the penetrance of the hearing phenotype in the *Slc7a8$^{-/-}$* mice is incomplete (*Figure 1D and E*). Therefore, mice were classified according to their hearing loss (HL) phenotype, defining normal hearing when ABR thresholds for all frequencies were <45 dB SPL, mild phenotype when at least two thresholds were between 45 and 60 dB SPL and severe hypoacusis when at least two thresholds were >60 dB SPL. At 4–6 months of age, *Slc7a8$^{-/-}$* mice showed either

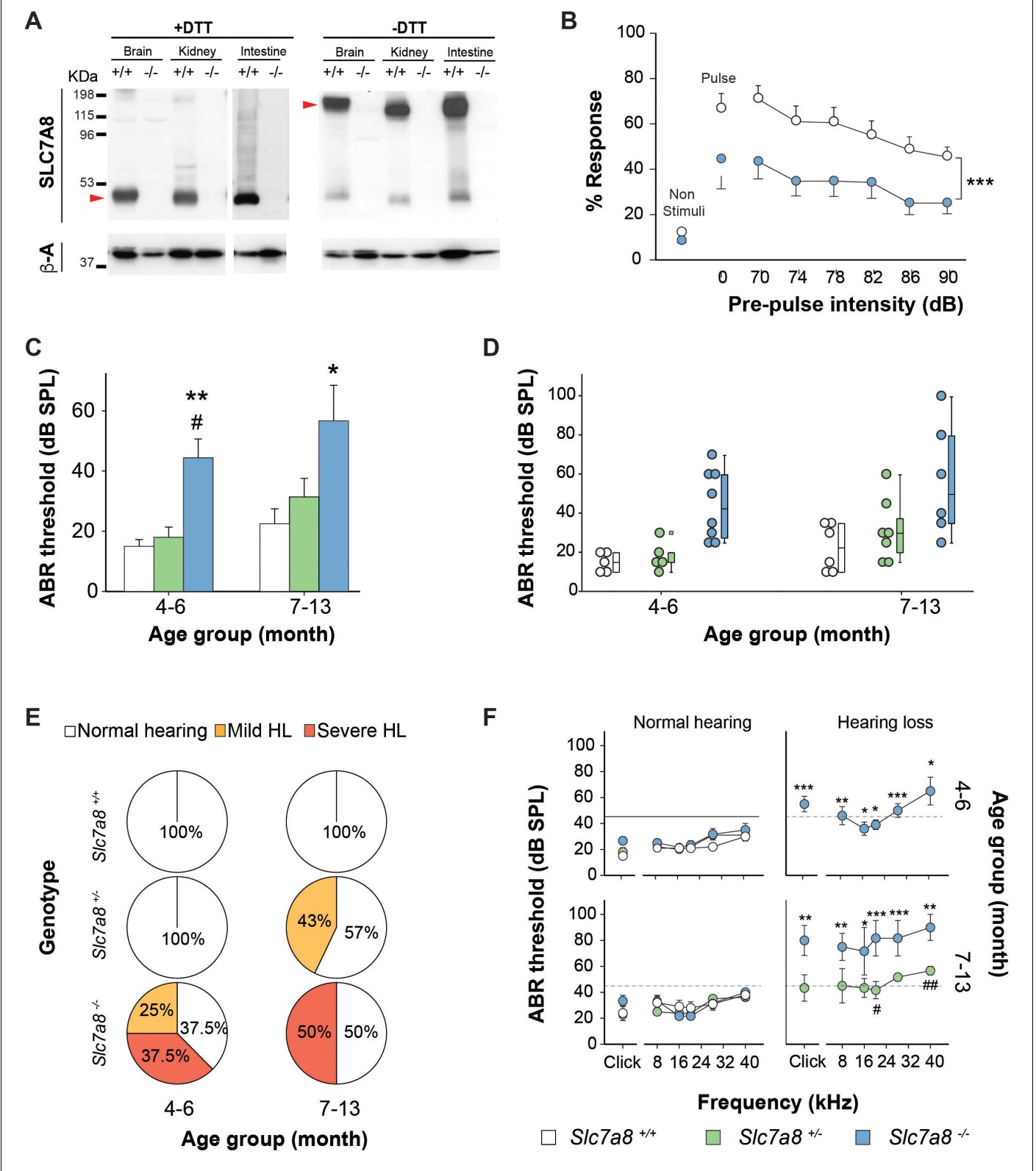

**Figure 1.** Hearing phenotype of C57BL6/J-129Sv *Slc7a8* knockout mice. (**A**) Representative image of western bloting of total membranes from kidney, brain and intestine of wild-type (+/+) and *Slc7a8* knock out (-/-) mice in the absence (-) or presence (+) of 100 mM dithiothreitol reducing agent (DTT) of three independent biological samples for both sexes (male and female). Protein (50 μg) were loaded in 7% acrylamide SDS-PAGE gel. Molecular mass standard (KDa) are indicated. Red arrows point SLC7A8/CD98hc heterodimer band as well as the light subunit SLC7A8. Upper panel: Rabbit anti-

*Figure 1 continued on next page*

*Figure 1 continued*

SLC7A8. Bottom panel: Mouse anti-βactin. (B) Pre-Pulse Inhibition of the acoustic startle response (PPI). Mean and SEM are represented. Pulse: 120 dB single pulse. Pre-pulse inhibition test: six different pre-pulse intensities (70 to 90 dB) in pseudo random order with 15 s inter-trial intervals. Wild type (white circles, n = 19) and *Slc7a8*$^{-/-}$ (blue circles, n = 15) from 4- to 7-month-old are represented. Significant differences were determined using paired Student's t-test, ***p<0.001 (C–F) Hearing phenotype in wild-type (*Slc7a8*$^{+/+}$, white, n = 11), heterozygous (*Slc7a8*$^{+/-}$, green, n = 12) and knockout (*Slc7a8*$^{-/-}$, blue, n = 14) mice, grouped by age (4–6 and 7–13 month old). (C,D) Auditory Brainstem Response (ABR) threshold in response to click, expressed as mean ±standard error (C), individual value (scatter plot, (D) and median (boxplot, (D). The significance of the differences was evaluated using ANOVA test, *p<0.05, **p<0.01 (*Slc7a8*$^{-/-}$ versus *Slc7a8*$^{+/+}$) and # p<0.05 (*Slc7a8*$^{-/-}$ versus *Slc7a8*$^{+/-}$). (E) Pie plot showing the percentage of normal hearing (all thresholds <45 dB SPL, white) mice and mice with mild (at least two tone burst threshold >45 dB SPL, orange) and severe (at least two tone burst threshold >60 dB SPL, red) hearing loss (HL), within each genotype and age group. (F) ABR thresholds in response to click and tone burst stimuli (8, 16, 24, 32 and 40 kHz) in mice from three genotypes separated by age group and hearing phenotype (normal hearing or hearing loss). Significant differences were determined using ANOVA test, *p<0.05, **p<0.01, ***p<0.001 (hearing impaired *Slc7a8*$^{-/-}$ versus normal hearing *Slc7a8*$^{+/+}$) and # p<0.05 (hearing impaired *Slc7a8*$^{-/-}$ versus *Slc7a8*$^{+/-}$).

DOI: https://doi.org/10.7554/eLife.31511.003

The following figure supplements are available for figure 1:

**Figure supplement 1.** Scheme of *Slc7a8* knockout mouse generation.
DOI: https://doi.org/10.7554/eLife.31511.004
**Figure supplement 2.** SLC7A8 expression in mouse brain.
DOI: https://doi.org/10.7554/eLife.31511.005
**Figure supplement 3.** Behavior phenotype.
DOI: https://doi.org/10.7554/eLife.31511.006
**Figure supplement 4.** ABR latencies and amplitudes of C57BL6/J-129Sv *Slc7a8* knockout mice.
DOI: https://doi.org/10.7554/eLife.31511.007
**Figure supplement 5.** Hearing phenotype of C57BL/6J *Slc7a8* knockout mice.
DOI: https://doi.org/10.7554/eLife.31511.008

severe (37.5%) or mild (25%) hearing loss, whilst mice from the other genotypic groups did not show hearing loss (*Figure 1E*). Next we studied 7–13 month-old mice, 50% of *Slc7a8*$^{-/-}$ mice presented severe hypoacusis and the hearing loss spread to lower frequencies with age. *Slc7a8*$^{-/-}$ mice with hearing loss showed statistically significant differences in ABR parameters when compared to the other genotypes (*Figure 1F*). Moreover, 43% of *Slc7a8*$^{+/-}$ mice developed mild hearing loss at 7–13 months, whereas the age-matched wild-type mice maintained intact hearing indicating a predisposition toward hearing loss in aged *Slc7a8*$^{+/-}$ mice (*Figure 1E*).

The onset and severity of ARHL is attributed to both environmental and genetic factors (*Cruickshanks et al., 2010*). As the environmental factors were well controlled in all the experiments, thus the phenotypic variability could be attributed as the consequence of individual genetic differences. Indeed, it has been described that several strains of inbred mice present a predisposition to suffer ARHL dependent on multiple genetic factors (*Kane et al., 2012*; *Murillo-Cuesta et al., 2010*). Here, the hearing loss phenotype was confirmed in a second mouse strain, the inbread C57BL6/J genetic background (*Figure 1—figure supplement 5*). Additionally, longitudinal study of *Slc7a8*$^{-/-}$ mice into the inbred C57BL6/J genetic background showed higher penetrance than the mixed background throughout the ages studied (*Figure 2—figure supplement 2*).

## Localization and quantification of SLC7A8 in the inner ear

The presence of SLC7A8 has previously been reported in the mouse cochlea (*Yang et al., 2011*; *Uetsuka et al., 2015*; *Sharlin et al., 2011*), and specifically localized to the stria vascularis by liquid chromatography tandem mass spectrophotometry and by Western blotting (*Uetsuka et al., 2015*). Here, SLC7A8 was detected in wild-type mouse cochlea by immunofluorescence supporting its localization to the spiral ligament and spiral limbus from the basal to the apical regions of the cochlea (*Figure 2A and B*). SLC7A8 immunolabeling was not observed in the stria vascularis. We observed an intense expression of SLC7A8 in the spiral ligament surrounding the stria indicating that the SLC7A8 epitope (*Figure 1—figure supplement 1B*) is either hidden or absent in the stria vascularis. Quantification of SLC7A8 expression in the cochlea showed half a dose of the transporter in the *Slc7a8*$^{+/-}$ than in wild-type mice, and its ablation in *Slc7a8*$^{-/-}$ mice (*Figure 2C*). A closer study of SLC7A8 immunofluorescence showed that the transporter is also expressed in the spiral ganglia neurons area (SGN) (*Figure 1—figure supplement 2B*).

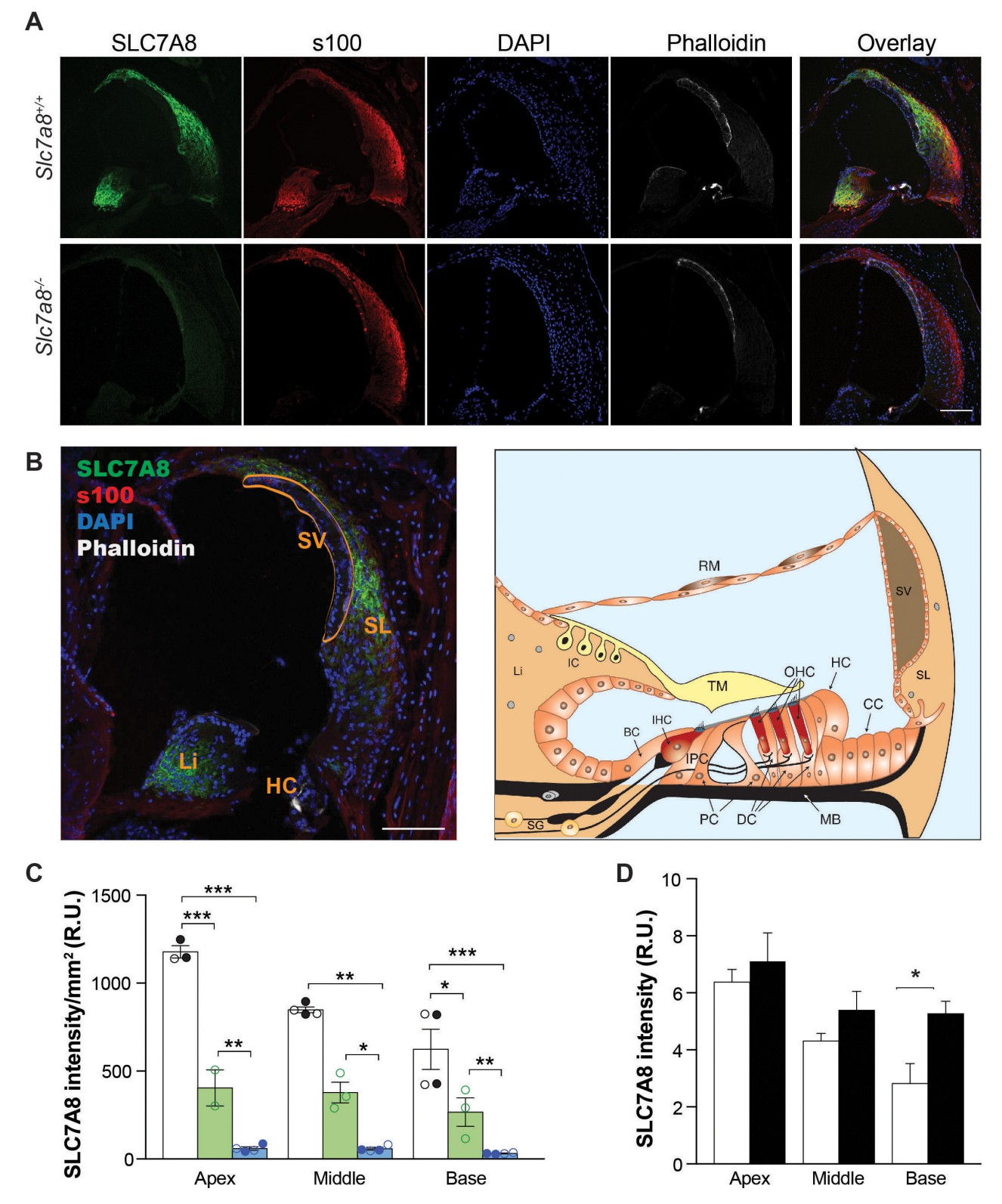

**Figure 2.** Immunolocalization of SLC7A8 in the mouse cochlea. (A) Representative photomicrographs of cryosections of the base of the cochlea showing immunodetection for SLC7A8 (green) and s100 (red); and staining for DAPI (blue) or phalloidin (white) of wild type (upper row) and *Slc7a8⁻/⁻* mice (lower row). Scale bar, 100 µm. (B) On the left overlay image of a wild-type section indicating cochlea areas. Scale bar, 100 µm. On the right schematic drawing of the adult scala media adapted from *Sanchez-Calderon et al. (2010)*. BC, border cells; CC, Claudius's cells; DC, Deiter's cells;

*Figure 2 continued on next page*

*Figure 2 continued*

HC, Hensen's cells; IC, intermediate cells; IHC, inner hair cells; IPC, inner phalangeal cells; Li, spiral limbus; MB, Basilar Membrane; OHC, outer hair cells; PC, pillar cells; RM, Reisner's membrane; SG, spiral ganglion; SL, spiral ligament; SV, stria vascularis; TM, tectorial membrane. (C) Quantification of SLC7A8 expression. Intensity of SLC7A8 immunofluorescence was normalized per mm$^2$. Mean ±SEM from quadruplicates for each section, taken from apex, middle and basal cochlear turns of 4 wild-type (black), 3 $Slc7a8^{+/-}$ (green) and 4 $Slc7a8^{-/-}$ (blue) young (4- to 7-month-old) mice. Open and closed circles represent individual mice from C57BL6/J-129Sv or C57BL6/J backgrounds, respectively. Unpaired Student's t-test statistical analysis, p-values: *,$\leq$0.05; **,$\leq$0.01 and ***,$\leq$0.001. (D) Quantification of SLC7A8 protein expression in the apex, middle and basal cochlear turns normalized per nuclei of young (2 month-old) (open bars) and old (12 month-old) (black bars) wild-type CBA mice. Data (mean ±SEM) were obtained from four cochlear sections obtained from three mice per group. Unpaired Student's t-test statistical analysis, p-value: *,$\leq$0.05.

DOI: https://doi.org/10.7554/eLife.31511.009

The following figure supplements are available for figure 2:

**Figure supplement 1.** Quantification of transcripts in the $Slc7a8^{-/-}$ mouse cochlea.

DOI: https://doi.org/10.7554/eLife.31511.010

**Figure supplement 2.** Progression of hearing phenotype of C57BL/6J $Slc7a8$ knockout mice.

DOI: https://doi.org/10.7554/eLife.31511.011

The early HL onset and the progressive ARHL phenotype observed in $Slc7a8^{-/-}$ and $Slc7a8^{+/-}$ mice respectively, prompted us to compare the expression of SLC7A8 in wild-type cochlea at different ages (*Figure 1D*). Immunofluorescence quantification of SLC7A8 intensity at 2- and 12 months of age showed expression in the young mice and increased presence of the transporter in the older mice (*Figure 2D*). In the same line, $Slc7a8$ mRNA quantification from cochlea extracts showed a progressive increased expression throughout mouse life (*Figure 2—figure supplement 1A*).

## Lack of $Slc7a8$ induced damage in the organ of Corti, spiral ganglion and stria vascularis

The cytoarchitecture of the inner ear was studied by hematoxylin/eosin staining (*Figure 3*), immunofluorescence (*Figure 4* and *Figure 4—figure supplement 1*) and mRNA detection of several cochlear markers (*Figure 3D* and *Figure 2—figure supplement 1*). Most of the structures of the cochlear duct, including spiral ligament, spiral limbus, tectorial and basilar membranes showed a normal gross cytoarchitecture in the $Slc7a8^{-/-}$ mice. In contrast, in the basal turns of the cochlea we observed that 3 out of 6 $Slc7a8^{-/-}$ mice evaluated showed complete loss of hair cells and flat epithelia, while only one $Slc7a8^{-/-}$ mouse showed intact epithelia in the organ of Corti (*Figure 3A*). Likewise, loss of cells in the spiral ganglia, especially in the basal regions of the cochlea, was observed (*Figure 3A*). $Slc7a8^{-/-}$ mice at 4 to 7 months of age presented ~50% of cell loss in the spiral ganglion compared with wild type mice (*Figure 3B*). Decreased number of cells in the ganglia significantly correlates with ABR threshold and HL phenotype (*Figure 3—figure supplement 1and B*). Concomitantly with the loss of hair cells and spiral ganglion (SG) nuclei in $Slc7a8^{-/-}$ mice, the messenger levels of cell type specific biomarkers, such as the potassium voltage-gated channels $Kcnq2$, $Kcnq3$ and $Kcnq5$, and the transporter $Slc26a5$, which are expressed in the organ of Corti and SG were down-regulated respectively (*Figure 3D* and *Figure 2—figure supplement 2B*).

Less densely packed cells in the spiral ligament were observed in $Slc7a8^{-/-}$ than in wild-type mice (*Figure 3A*). Reinforcing this observation, the expression of Kir4.1, a potassium channel highly expressed in stria vascularis cells (*Ando and Takeuchi, 1999*), was also dramatically reduced by 50% in $Slc7a8^{-/-}$ (*Figure 4B* and *Figure 4—figure supplement 1A*). Likewise, decreased expression of Kir4.1 marker correlates with HL phenotype (*Figure 3—figure supplement 1C*). Phalloidin labeling of actin fibers in the basal cells of the stria vascularis was also decreased 50% in the base of the cochlea (*Figure 4C* and *Figure 4—figure supplement 1*).

SLC7A8 is abundantly expressed in fibrocytes of the spiral ligament and limbus (*Figure 2*), accordingly the number of fibrocytes in the spiral ligament decreased by 2/3 and 1/3 in the null and $Slc7a8^{+/-}$ mice, respectively (*Figure 3C*). Moreover, mice with severe HL phenotype showed 30% less number of fibrocytes in the spiral ligament (*Figure 3—figure supplement 1D*). The expression of the transcription factor $Tbx18$, essential for fibrocytes development and differentiation, was 50% less in $Slc7a8^{-/-}$ than in wild-type mouse cochleae (*Figure 3D*). In contrast, the expression of s100, fibrocyte types I and II marker, did not show significant differences (*Figure 4D* and *Figure 4—figure supplement 1C*).

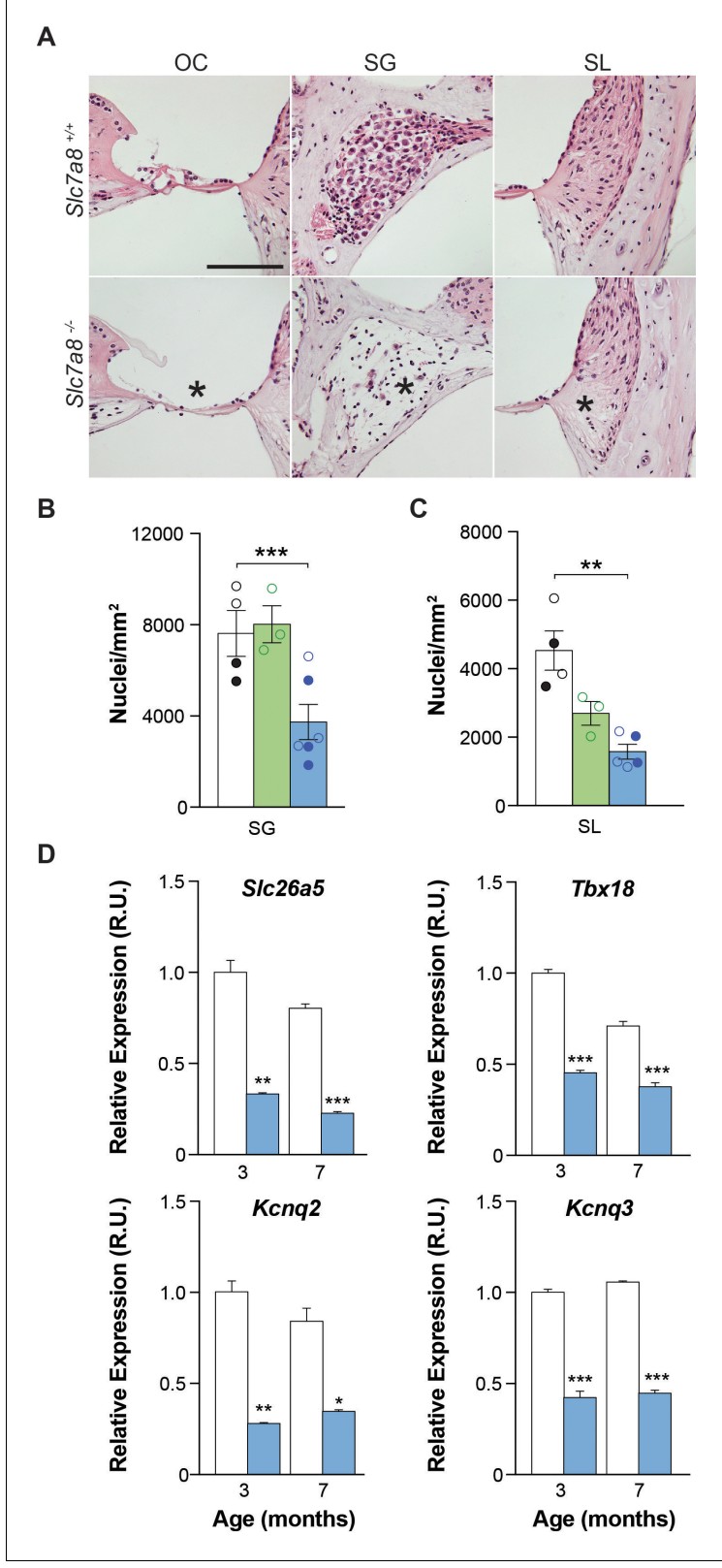

**Figure 3.** Cytoarchitecture of the *Slc7a8*$^{-/-}$ mouse cochlea. (**A**) Hematoxylin and Eosin staining of the base of the cochlea. Representative photomicrographs taken from paraffin sections of wild-type and hipoacusic *Slc7a8*$^{-/-}$ mice. OC, Organ of Corti; SG, spiral ganglia region; and SL, spiral ligament. * Indicates loss of hair cells in the organ of Corti (first column), loss of neurons in the spiral ganglia (second column) and lower nuclei density in the

*Figure 3 continued on next page*

*Figure 3 continued*

spiral ligament (third column). Scale bar 100 μm. (**B**) Quantification of the number of neurons in the spiral ganglia (SG) in the basal turns of the cochlea. Y axis represents the mean nuclei quantification of 5 to 10 areas in SG. (**C**) Quantification of the number of nuclei in the spiral ligament (SL) of the basal turns of the cochlea by immunofluorescence using DAPI staining. For each sample, 12 overlaps of Z-stacks areas were used to quantify number of nuclei. Unpaired Student's t-test statistical analysis: **, p≤0.01 (**A to C**) 4 wild-type (black), 3 *Slc7a8+/−* (green) and 4 *Slc7a8−/−* (blue) mice at 4 to 7-month-old are represented. Circles represent the average of the quadruplicate analysis performed in each mouse of C57BL6/J-129Sv (open) and C57BL6/J (filled) background. (**D**) Quantification of mRNA markers by RT-qPCR PCR. Cochlear gene expression of *Slc26a5, Tbx18, Kcnq2* and *Kcnq3* in the cochlea at 3-month-old and 7 months wild-type (white bars) and *Slc7a8−/−* (blue bars) C57BL6/J mice. Expression levels, normalized with *Rplp0* gene expression, are represented as n-fold relative to control group. Values are presented as mean ±SEM of triplicates from pool samples of three mice per condition. Unpaired Student's t-test statistical analysis, p-values: *p≤0.05; **p≤0.01; ***p≤0.001.
DOI: https://doi.org/10.7554/eLife.31511.012
The following figure supplement is available for figure 3:

**Figure supplement 1.** Correlations of the cell numbers and cell type biomarkers with HL phenotype.
DOI: https://doi.org/10.7554/eLife.31511.013

## Mutations in *SLC7A8* are associated with ARHL

Once we associated mouse SLC7A8 transporter with deafness and identified it as a potential ARHL gene, screening for mutations in human populations was initiated. Whole genome sequencing (WGS) and audiogram test data obtained from 147 individuals from isolated villages in Italy were included in the study. The inclusion criteria were people 50 years old or older with an audiogram test done at high frequencies (Pure-tone audiometric PTA-H, 4 and 8 kHz). Individuals with pure-tone average for high frequencies (PTA-H) greater than or equal to 40 decibels hearing level (dB HL) were considered ARHL cases, whilst people with PTA-H less than 25 dB were considered as controls. A total of 66 cases suffering ARHL and 81 controls were selected. The gene-targeted studies conducted in this isolated cohort succeeded in detecting seven heterozygous missense variants (*Table 1*). Four of the variants: p.Val460Glu (V460E), p.Thr402Met (T402M), p.Val302Ile (V302I) and p.Arg418His (R418C) belong to ARHL cases (see Audiogram in *Figure 5—figure supplement 1A*) and other three: p.Arg8Pro (R8P), p.Ala94Thr (A94T) and p.Arg185Gln (R185L) to the control group (see Audiogram in Audiogram in *Figure 5—figure supplement 1B*).

All the mutations found in *SLC7A8* cases and controls from isolated villages of Friuli Venezia Giulia exhibited different frequencies in comparison to public data bases, such as ExAC among others (see *Table 1*). According to ExAC database's constrain metrics (*Lek et al., 2016*), the gene shows evidence of tolerance of both loss of function (pLi = 0) and missense variation (missense Z score = −0.14).

## Functional studies of *SLC7A8* mutations

A structural model of human SLC7A8 protein built using the homologous protein AdiC (*Kowalczyk et al., 2011*) in the outward-facing conformation (*Rosell et al., 2014*) (*Figure 5—figure supplement 1C and D*) was used to localize all the mutations identified here. Interestingly, three of the four mutations found in ARHL patients were located in very striking places: (i) V302 is a conserved amino acid located in the extracellular loop four which corresponds to the external lid that closes the substrate binding site when the transporter is open to the cytosol, (ii) T402 is located in transmembrane (TM) domain 10 facing to the substrate binding site, and (iii) V460 is located at the very end of TM domain 12, with potential interaction with the plasma membrane. In contrast, R418 is in the intracellular loop 5, between TM domain 10 and TM domain 11 and with no functional role described in transporters with the LeuT-fold (*Krishnamurthy and Gouaux, 2012*). Thus, three of these mutations were promising candidates to affect the transporter function due to their crucial location.

In vitro functional characterization of variants present in patients with ARHL and controls was performed by measuring amino acid uptake in HeLa cells co-transfected with the heavy subunit CD98hc and Strep tagged-SLC7A8 wild type and variants (*Figure 5*). Co-expression of the light (SLC7A8) and the heavy (CD98hc) subunits in the same cell increases the plasma membrane localization of the

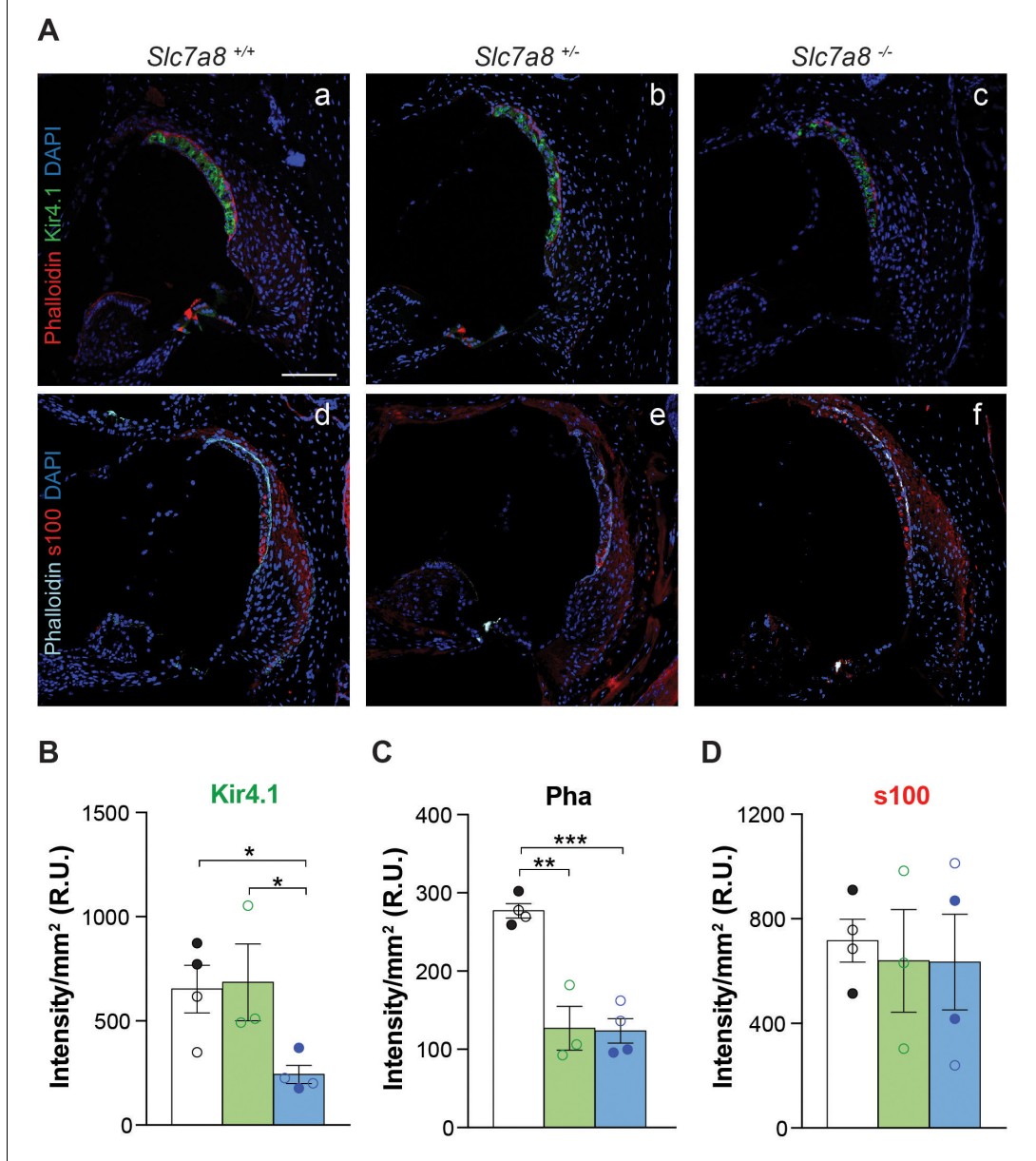

**Figure 4.** Immunofluorescence of cochlear markers in the *Slc7a8*$^{-/-}$ mouse. (**A**) Representative photomicrographs of cryosections (10 μm) from the basal turn of the cochlea from wild type (1 and 4), *Slc7a8*$^{+/-}$ (2 and 5) and *Slc7a8*$^{-/-}$ (3 and 6) mice labeled for Kir4.1 (green), phalloidin (red) and DAPI (blue) (1 to 3), or for s100 (red), phalloidin (cyan) and DAPI (blue) (4 to 6). Scale bar, 100 μm. (**B, C and D**) Graph representing the quantification of Kir4.1, s100 and phalloidin (Pha) labeling intensity in the basal turn of the cochlea. Means ± SEM, normalized per mm$^2$ of 4 wild type (black bars), 3 *Slc7a8*$^{+/-}$ (green bars) and 4 *Slc7a8*$^{-/-}$ (blue bars) young (4- to 7-month-old) mice are represented. Individual circles represent the average of the quadruplicate analysis of sections from each mice of either C57BL6/J-129Sv (open) or C57BL6/J (filled) backgrounds. Unpaired Student's t-test statistical analysis, p-value: *,≤0.05.
DOI: https://doi.org/10.7554/eLife.31511.014

The following figure supplement is available for figure 4:

**Figure supplement 1.** Quantification of the intensity of cell type biomarkers in apical and middle cochlear regions.
DOI: https://doi.org/10.7554/eLife.31511.015

Table 1. *SLC7A8* Humans mutations found in ARHL and controls individuals.

| Phenotype | Age | Sex | Chr. 14 | Variant | Consequence | Code | Frequency | | | | | Studied cohort |
|---|---|---|---|---|---|---|---|---|---|---|---|---|
| | | | | | | | Esp6500siv2 | 1000 g | Campion | ExAC | | |
| ARHL | 75 | Female | 23597290 | 14:23597290 A / T | p.Val460Glu | V460E | NA | NA | 0.0013 | 0.00002475 | | 0.015 |
| ARHL | 57 | Male | 23598917 | 14:23598917 G / A | p.Thr402Met | T402M | NA | NA | 0.0047 | 0.00002471 | | 0.015 |
| ARHL | 75 | Male | 23608641 | 14:23608641 C / T (rs142951280) | p.Val302Ile | V302I | 0.0005 | NA | 0.0047 | 0.0004613 | | 0.015 |
| ARHL | 86 | Female | 23598870 | 14:23598870 G / A (rs146946494) | p.Arg418Cys | R418C | 0.0005 | NA | 0.002 | 0.00002477 | | 0.015 |
| control | 50 | Male | 23652101 | 14:23652101 C / G (rs141772308) | p.Arg8Pro | R8P | 0.0008 | NA | 0.0013 | 0.0008156 | | 0.012 |
| control | 50 | Male | 23635621 | 14:23635621 C / T (rs139927895) | p.Ala94Thr | A94T | 0.0012 | 0.002 | 0.0013 | 0.00202 | | 0.012 |
| control | 90 | Female | 23612368 | 14:23612368 C / A (rs149245114) | p.Arg185Gln | R185L | NA | NA | 0.002 | 0.00002471 | | 0.012 |

ARHL (age-related hearing loss). The age (years) of the subject when the Audiogram was performed is indicated. Variant [CHR: position reference/alternate (dbSNP135rsID)]. Consequence [HGUS annotation (protein change)]. Code [short description of the alternate variant]. Frequency of the mutations: Esp6500siv2 (NHLBI Exome Sequencing Project), 1000 g (1000 Genomes Project), Campion (The Allele Frequency Net Database) and ExAC (The Exome Aggregation Consortium).

DOI: https://doi.org/10.7554/eLife.31511.016

transporter (*Rosell et al., 2014*). All tested variants showed expression levels comparable to those of wild type, except for V460E that showed only 20% expression of wild -ype protein (*Figure 5— source data 1*), being the only variant that did not reach the plasma membrane as indicated by the lack of co-localization with wheat germ agglutinin staining (*Figure 5A*). Amino acid transport induced by SLC7A8 was analyzed for wild type and the identified variants (*Figure 5B*). All variants present in controls (R8P, R186L and A94T) conserved more than 80% of alanine transport compared with wild-type protein. Three variants found in patients with ARHL showed diminished alanine transport activity: T402M and V460E presented little residual transport activity (14.6 ± 2.6% and 3.6 ± 0.3% of wild-type activity, respectively) and R418C showed 50.7 ± 5.4% of wild-type alanine transport. Surprisingly, V302I presented similar alanine transport levels to wild type SLC7A8. Location of residue V302 within EL4 (within the external substrate lid (*Figure 5—figure supplement 1D*) led us to additionally measure a larger size SLC7A8 substrate, whose transport could potentially be more compromised than that of a small substrate (e.g. alanine). Interestingly, V302I transport activity of tyrosine was found to be only 40.0 ± 1.6% of wild-type SLC7A8. Because the V302I mutation showed a substrate-dependent impact, tyrosine transport in the other variants was also tested (*Figure 5B*). Other SLC7A8 variants found in patients with ARHL and controls showed similar decreased transport activity for alanine and tyrosine. Thus, the SLC7A8-induced tyrosine transport was clearly defective in the four variants found in patients with ARHL, whereas it was barely affected (>85% of wild-type transport activity) in the variants found in controls.

## Discussion

Here, we show that loss of function of the amino acid transporter SLC7A8 is associated with ARHL in both humans and mice. Full ablation of SLC7A8 transporter in mice produced a hearing loss defect with incomplete penetrance affecting mainly high-frequency sounds, a characteristic of ARHL (*Figures 1C–F*, S5 and S6). Interestingly, hearing loss severity increases with age in *Slc7a8*$^{-/-}$ mice (*Figures 1C–F* and S6). Similarly, *Slc7a8* heterozygous mice showed increased hearing loss penetrance with age, as indicated by the late onset of the phenotype (starting from 7 months onwards) (*Figures 1E*, S5 and S6). In addition, SLC7A8 expression in wild type cochlea rises during ageing (*Figure 2D* and S7A). In patients with ARHL we identified four SLC7A8 variants that showed loss of function of transport of tyrosine (*Figure 5B*). Altogether, these results indicate that full SLC7A8

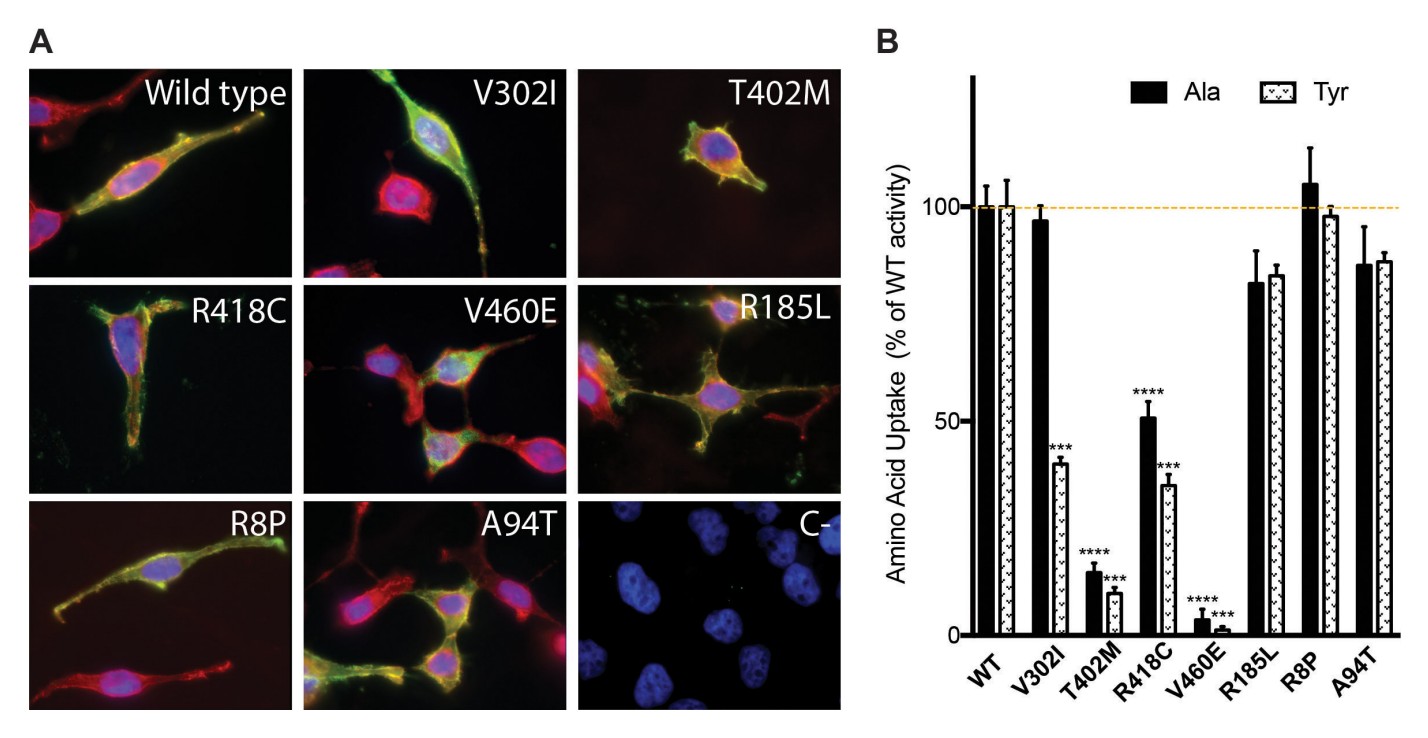

**Figure 5.** In vitro characterization of SLC7A8 mutants. (**A**) Panel showing representative images of immunofluorescence of wild type and the indicated SLC7A8 mutants overexpressed in HeLa cells. Overlay of SLC7A8 (green), wheat germ agglutinin (WGA, membrane marker) (red) and the nuclear marker DAPI (blue) labeling. All SLC7A8 variants, except V460E, reached the plasma membrane. (**B**) Alanine (Ala) and tyrosine (Tyr) transport activity of human SLC7A8 wild type (WT) and mutants in transfected HeLa cells. SLC7A8 transport activity, corrected by SLC7A8-GFP expression, is presented as percentage of wild-type SLC7A8 transport activity. Data (mean ±SEM) corresponds to three independent experiments with quadruplicates. Mutants activity comparing with its, respectively, wild-type transport unpaired Student's t-test statistical analysis is represented, p-values: *,$\leq$0.05; **,$\leq$0.01 and ***,$\leq$0.001.

DOI: https://doi.org/10.7554/eLife.31511.017

The following source data and figure supplement are available for figure 5:

**Source data 1.** Mutants expression and oligonucleotides for Site-Directed Mutagenesis (5'-3').
DOI: https://doi.org/10.7554/eLife.31511.019

**Figure supplement 1.** Audiogram of patients with ARHL and localization of the mutations in SLC7A8 protein.
DOI: https://doi.org/10.7554/eLife.31511.018

function is needed to keep an optimal hearing function throughout life, with half a dose of SLC7A8 being enough to accelerate ARHL phenotype in mice and humans.

The hearing loss (HL) phenotype in the $Slc7a8^{-/-}$ mice has been confirmed on two genetic backgrounds (mixed C57BL6/J-129Sv; *Figure 1*, and inbred C57BL6/J; *Figure 1—figure supplement 5*). Interestingly, onset and penetrance, but not severity, was increased in the hearing loss trait of $Slc7a8^{-/-}$ mice in the pure C57BL6/J background (*Figure 1—figure supplement 5*). It is well-known that the C57BL6/J background carry a mutation in the *Cdh23* gene causing early onset of ARHL (*Noben-Trauth et al., 2003*; *Mazelová et al., 2003*). It is also worthwhile to mention that all the inbred C57BL6/J mice used to perform the experiments in this research were positive for the ARHL susceptibility allele A/A in *Cadh23* (data not shown). Genetic linkage between both genes could be disregarded because both are located in different chromosomes (*Slc7a8* in Chr:14 and *Cdh23* in Chr:8). Therefore, non-additive severity of the hearing loss phenotype of *Slc7a8* ablation and *Cdh23* susceptibility allele suggests that both genes may share similar mechanisms of pathogenicity.

In line with the results observed in the mouse model, the four human mutations found in heterozygosis in ARHL patients showed a reduced SLC7A8 transporter activity meanwhile the mutations found in control group did not affect the transporter activity (*Figure 5B*). The predisposition of *SLC7A8* to host deleterious variants, as shown by the in silico-patterns of missense and loss-of-

function tolerance, could be explained because its aberration affects age-related hearing function, but its ablation is neither vital nor affects the reproduction of the mice (Slc7a8$^{-/-}$ showed same frequency of siblings as expected, data not shown). Furthermore, the presence of mutations in both ARHL cases and controls in our cohort with higher frequencies in respect to public databases could be explained as a result of isolation and inbreeding in our individuals; as isolation in a population could lead to an enrichment of deleterious variants due to relaxation of purifying selection (*Xue et al., 2017*). We also noted that in ExAC the mutations found in controls have a mean frequency that is seven times higher than the ones found in our cases, and we speculate that this could be an indirect hint of the higher deleteriousness of the variations found in our cases in respect to the controls. Thus, the present work points to *SLC7A8* as a strong candidate gene involved in ARHL induction and the presented data suggest that a significant proportion (~3%) of ARHL cases could be explained by SLC7A8 mutations making it one of the major players so far described.

SLC7A8 was localized in key cochlear structures: the spiral ligament, spiral limbus and spiral ganglion (*Figure 2A* and S2B) likewise the three main pathological changes described in the ARHL were observed in the absence of SLC7A8: the hair cells of organ of Corti (sensory), the spiral ganglia (neural), and the spiral ligament and the stria vascularis (metabolic) (*Figure 3*).

The spiral ligament contributes to cochlear homeostasis and is crucial for normal hearing. Degradation of the spiral ligament can result in either one form of hereditary deafness through *POU3F4* mutations at locus DFN3 (41) or in the loss of endocochlear potential (EP) in presbycusis mouse models (*Wu and Marcus, 2003*). In the spiral ligament, SLC7A8 expression was detected in fibrocytes, mostly in type I, close to the stria vascularis (*Figure 2*). In addition, a reduced number of total cells was observed in both Slc7a8$^{-/-}$ and Slc7a8$^{+/-}$ mice (*Figure 3C*). Type I fibrocytes are interconnected with the adjacent types II and V cells forming a gap junction-dependent cell system with a relevant role in ion homeostasis [for a review, see (*Kikuchi et al., 2000*)]. Deafness due to fibrocyte alterations has been described, which indicates the importance of their integrity for appropriate hearing (*Minowa et al., 1999*; *Teubner et al., 2003*; *Boettger et al., 2003*; *Delprat et al., 2005*; *Trowe et al., 2008*). Nonetheless, s100 expression (*Figure 4C*) appeared to be unaffected in the absence of SLC7A8. Interestingly, mutations in genes expressed in spiral ligament fibrocytes could affect stria vascularis function causing deafness, such as the ablation of the fibrocyte transcription factor *POU3F4* that causes loss of fibrocytes IV and V in the spiral ligament, decreased cellular density in the stria vascularis and decreased expression of Kir4.1 (48). As the stria vascularis regulates nutrient transport and ion fluxes is responsible for the maintenance of the EP (*Peter and Santi, 2001*), which is the driving force required for neurotransmission after acoustic stimulus (*Wangemann, 2006*; *Couloigner et al., 2006*). We observed alterations in the stria vascularis, decreased expression of Kir4.1 and the basal cell marker phalloidin all correlating with HL phenotype in Slc7a8$^{-/-}$, and similar traits in Slc7a8$^{+/-}$ mice (*Figures 3A* and *4B–C* and S9). Moreover, is described that the ablation of the T-box transcription factor gene *Tbx18*, expressed in the spiral ligament, compromises fibrocytes differentiation (*Trowe et al., 2008*) and concomitant disruption of the architecture of the stria vascularis with almost complete absence of the basal cell layer, and down-regulation of Kir4.1 (52). Likewise, deletion of Pendrin (*SLC26A4*, PDS) (Cl$^-$/I$^-$/HCO3$^-$ anion exchanger expressed in mouse fibrocytes) showed pronounced signs of vestibular disease attributed to an altered EP (*Everett et al., 2001*). Concomitant with reported data, transcript levels of both *Tbx18* and *Slc26a4* are down-regulated in the Slc7a8$^{-/-}$ mouse (*Figure 3D*). Therefore, if we assume a defect in ion homeostasis in the absence of SLC7A8, we could expect an EP impairment that should also trigger vestibular damage. In line with this assumption, we observed impaired balance during gradual acceleration in rotarod test performance of Slc7a8$^{-/-}$ mouse (*Figure 1—figure supplement 3G*). Altogether, the data presented suggests that the absence of SLC7A8 in fibrocytes might contribute a metabolic component to the progression of hearing loss.

The reduction in the number of cells of the spiral ganglia in Slc7a8$^{-/-}$ mice to half of those in wild type (*Figure 3A and B*) and its correlation with ABR threshold at high frequencies (*Figure 3—figure supplement 1*) could be considered causative of neuronal hearing loss (*Camarero et al., 2001*), and the lack of expression of SLC7A8 in SG might directly contributed to this neurodegeneration (*Figure 1—figure supplement 2B*). SG axons are part of the auditory nerve and transmit signals from the organ of Corti to the brain. In addition, it has been described that SG degeneration may result in hair cells and sensory hearing loss (*Stankovic et al., 2004*; *Sugawara et al., 2007*; *Zilberstein et al., 2012*). SLC7A8 is expressed in the SG but not in the organ of Corti. However,

*Slc7a8⁻/⁻* mice also showed loss of hair cells (*Figure 3A*) suggesting a potential negative feedback from the damaged SG similar to those described (*Stankovic et al., 2004*; *Sugawara et al., 2007*; *Zilberstein et al., 2012*).

SLC7A8/SLC3A2 exchanges all neutral amino acids except for proline (*Pineda et al., 1999*), and therefore either SLC7A8 ablation in mice or *SLC7A8* loss-of-function mutations in humans can alter availability or concentration of a specific set of neutral amino acids in cells (especially fibrocytes and neurons) of the spiral ligament, spiral limbus and spiral ganglion. Three of the four ARHL mutations (T402M, R418C and V460E) showed similarly compromised transport of the amino acids tested (alanine and tyrosine), whereas V302I selectively showed a defect for the large amino acid tyrosine (*Figure 5B*). Mutation V302I, located within the external lid in the extracellular loop 4, might result in a steric hindrance with bulky substrates when closing the substrate cavity in the inward-facing conformation of the transporter. SLC7A8 loss-of-function might render alterations in the cell content of bulky neutral amino acids like branched chain amino acids or glutamine, which affect proteostasis and renewal of cell structures causing cell stress (*Efeyan et al., 2015*; *Someya and Prolla, 2010*). Caloric restriction, that involves both an increased branched amino content and protein degradation, showed an effective delay of age-related cochlear neuron degeneration (*Someya et al., 2010*; *Bao and Ohlemiller, 2010*). In any case, *Slc7a8⁻/⁻* cochlea presents signs of unresolved chronic inflammation with up-regulation of *Il1b* and *Il6* mRNA (*Figure 2—figure supplement 1C*) and reduced activation of macrophages (down-regulation of Iba1 protein) (Suppl. Figure S8D). As SLC7A8 is also expressed in macrophages (BioGPS [Internet]. 2001), the role of the immune response in the hearing loss associated with *Slc7a8⁻/⁻* mice deserves further attention.

SLC7A8 also transports thyroid hormones (TH) (*Zevenbergen et al., 2015*; *Hinz et al., 2017*) as well as the dopamine precursor L-DOPA (*Gomes and Soares-da-Silva, 2002*; *Pinho et al., 2004*). Even though hypothyroidism causes hearing loss characterized by alterations in cochlear development (*Peeters et al., 2015*) and L-DOPA showed a protective role for cochlea during aging (*Murillo-Cuesta et al., 2010*), *Slc7a8⁻/⁻* mice showed neither hypothyroidism (*Braun et al., 2011*) nor alterations in L-DOPA plasma levels (data not shown). The lack of SLC7A8 might be compensated by other transporters like the main TH transporter MCT8 (*Núñez et al., 2014*). Moreover, we cannot disregard a local impact of a shortage of L-DOPA in the cochlea, which could influence its maintenance, altering the protective role of this metabolite. Therefore, in the absence of SLC7A8, three elements could play a role in the hearing loss phenotype: neutral amino acids, thyroid hormones and/or L-DOPA. Characterization of new SLC7A8 mutations with substrate-dependent transport activity will be necessary to draw a definitive conclusion as to the molecular mechanism of the SLC7A8 substrates involved in ARHL.

## Conclusion

The present work provides evidence that the amino acid transporter SLC7A8/SLC3A2 has a direct role in age-related hearing-loss (ARHL). The ablation of SLC7A8 in a mouse model causes deafness with ARHL characteristics, defective audition at high-frequencies with early onset in homozygotes and progressive worsening in heterozygotes with age. Identification of rare variants in *SLC7A8* gene together with amino acid transport loss-of-function in ARHL patients supports the concept that this gene has a role in the auditory system in association with other genetic and/or environmental factors.

This study highlights amino acid transporters as new targets to study in largely uncharacterized hearing disorders. The description of *SLC7A8* as a novel gene involved in a complex trait such as ARHL demonstrates the importance of amino acid homeostasis in preserving auditory function and suggests that genetic screening should be extended to consider other amino acid transporters as potential new genes involved in cochlear dysfunction. Our results may enable the identification of individuals susceptible to developing ARHL, allowing for early treatment or prevention of the disease.

## Materials and methods

All key research resources described in this section are summarized in *Table 2*.

**Table 2.** Key resources table.

| Reagent type (species) or resource | Designation | Source or reference | Identifiers | Additional information |
|---|---|---|---|---|
| Antibody | SLC7A8 antibody | Custom made | NA | Anti-Rabbit peptide sequence: PIFKPTPVKDPDSEEQP WB: 1:1000, IHC: 1/5000 and IF:1/200 |
| | s100 | Sigma-Aldrich | Ref: S2532 | IF: 1/1000 |
| | Kir4.1 | Merck Millipore | Ref: AB5818 | IF: 1/200 |
| | BA1 | Abcam | Ref: ab5076 | IF: 1/200 |
| | Phalloidin | Thermo Fisher Scientific | Ref: A22287 | IF: 1/100 |
| | Donkey anti-Goat Alexa Fluor 546 | Thermo Fisher Scientific | Ref: A-11056 | IF: 1/300 |
| | Donkey anti-Rabbit Alexa Fluor 488 | Thermo Fisher Scientific | Ref: A-21206 | IF: 1/300 |
| | Goat anti-Mouse Alexa Fluor 546 | Thermo Fisher Scientific | Ref: A-11030 | IF: 1/300 |
| | Goat anti-Rabbit Alexa Fluor 488 | Thermo Fisher Scientific | Ref: A-11034 | IF: 1/300 |
| | WGA | Thermo Fisher Scientific | Ref: W21405 | labeled with Texas-Red IF: 1 mg/mL |
| | Anti-Strep Tag GT517 | Abcam | Ref: ab184224 | IF: 1/100 |
| | Goat-anti-mouse-FITC | Abcam | Ref: ab6785 | IF: 1/300 |
| Behavior | Rotarod | Panlab | Ref:LE8500 | |
| | Treadmill | Panlab | E8710MTS | |
| | Morris water maze | Panlab | SMART camera | circular tank (150 cm diameter, 100 cm high) |
| | PPI | Panlab | LE116 | |
| | Restrain stressor | Lab Research | Ref:G05 | |
| | ABR | Tucker Davis Technologies TDT | System 3 Evoked | |
| Mouse | C57BL6/J wild type | Harlam | Ref: 057 | C57BL/6JOlaHsd |
| | C57BL6/J wild type | Jackson laboratory | Ref: 000664/Black | |
| | *Slc7a8*[-/-] chimera | Genoway | Customized Model Development | Strategy *Figure 1—figure supplement 1* |
| Cell Line | HeLa | Sigma Aldrich | Ref: 93021013 | |
| Chemical compound, drug | DTT dithiothreitol | SigmaAldrich | Ref:D9779 | |
| | L- [$^3$H]-labeled alanine | Perkin Elmer | Ref: NET348250UC | 1 μCi/ml |
| | [3 hr]-tyrosine | Perkin Elmer | Ref: NET127250UC | 1 μCi/ml |
| Commercial assay or kit | Pierce BCA Protein Assay Kit | Thermo Scientific | Ref:23225 | |
| | ECL | GE Healthcare | Ref:RPN2232 | |
| | Corticosterone EIA kit | Enzo | Ref:ADI900097 | |

*Table 2 continued on next page*

*Table 2 continued*

| Reagent type (species) or resource | Designation | Source or reference | Identifiers | Additional information |
|---|---|---|---|---|
| | A + B conjugate | Vectastain | Ref: ABC kit | |
| | Rneasy | Qiagen | Ref: 74104 | |
| | High-capacity cDNA Reverse Transcription Kit | Applied Biosystems | Ref: 4368813 | |
| | TaqMan Gene Expression Assay | Applied Biosystems | potassium voltage-gated channel subfamily Q member 2 (*Kcnq2*) Mm00440080_m1; potassium voltage-gated channel subfamily Q member 3 (*Kcnq3*) Mm00548884_m1; potassium voltage-gated channel subfamily Q member 5 (*Kcnq5*) Mm01226041_m1; prestin (*Slc26a5*) Mm00446145_m1; T-box transcription factor TBX18 (*Tbx18*) Mm00470177_m1; interleukin one beta (*Il1b*) Mm00434228m1; interleukin 6 (*Il6*) Mm00446190m1; solute carrier family 7 (cationic amino acid transporter, y + system), member 8 (*Slc7a8*) Mm01318971m1 | |
| | QuikChange site-directed mutagenesis kit | Stratagene | Ref: 200524 | |
| Gene (human) | *SLC7A8* | NCBI | NM_012244.3 | Protein NP_036376.2 (535AA) |
| | *Slc7a8* | NCBI | NM_016972.2 | Protein NP_058668.1 (531AA) |
| Sequence-based reagent | whole genome sequence | Illumina | HiSeq 2000 | Data coverage was ranging from 4 to 10X |
| | Sanger sequencing | Life Technologies | 3500 Dx Genetic Analyzer | |
| | BigDye | Life Technologies | ABI PRISM 3.1 Big Dye terminator | |
| Software, algorithm | BioSig | Tucker Davis Technologies TDT | NA | |
| | Graph Pad Software | GraphPad Software, Inc | Prism 4 | https://www.graphpad.com/scientific-software/prism/ |
| | SeqMan Pro software | DNAstar | https://www.dnastar.com/t-seqmanpro.aspx | sequencing assembly and analysis |
| | Annotations tools | ANNOVAR | http://annovar.openbioinformatics.org/en/latest/ | functional annotation of genetic variants DOI: 10.1093 |
| | Genome Research | Bcftools | http://samtools.github.io/bcftools/ | |
| | SPSS 23.0 statistic software package | IBM | NA | https://www.ibm.com/analytics/data-science/predictive-analytics/spss-statistical-software |
| Transfected construct | *Slc7a8* construct | Agilent | Catalog #212205 | Resistances: Neomycin and thymidine kinase |
| | pcDNA3.1-StrepTag | ThermoFisher | Ref: V79020 | fused SLC7A8 or SLC3A2 |

DOI: https://doi.org/10.7554/eLife.31511.020

## Mouse protocols

Animal experimentation complied with the ARRIVE guidelines and was conducted in accordance with Spanish (RD 53/2013) and European (Directive 2010/63/EU) legislations. All protocols used in this study were reviewed and approved by the Institutional Animal Care and Use Committee at IDI-BELL in a facility accredited by the Association for the Assessment and Accreditation of Laboratory Animal Care International (AAELAC accredited facility, B900010). Mice procedures were done according with scientific, humane, and ethical principles. The studied mouse model did not show phenotype differences comparing male and female. Thus, to ensure that our research represents

both genders, the studies describes in this work were performed using both sexes equitably. The number of biological and experimental replicates is detailed in the legend of each figure.

## Mouse model

Generation of the null *Slc7a8* (*Slc7a8*$^{-/-}$) was done by gene disruption. A coding region that includes exon 1 of the *Slc7a8* gene was replaced for a neomycin resistance cassette by homologous recombination using a pBlueScript vector with two homologous arms (right: 6.1 kb and left: 2.3 kb) and two resistances (neomycin and thymidine kinase) in 5' region of the gene (*Figure 1—figure supplement 1A*). ES cells transfection and microinjection experiments were done by GenOway (Lyon-France). Chimera mouse was outcrossed with a wild-type C57BL6/J mouse to obtain first generation (F1) of *Slc7a8* heterozygous (*Slc7a8*$^{+/-}$) in a mixed C57BL6/J-129Sv background. Intercross of F1 resulted in the analyzed F2 generation, which contemplates the three genotypes: wild type, *Slc7a8*$^{+/-}$ and *Slc7a8*$^{-/-}$ knockout mice. The pure inbred genetic background was generated backcrossing *Slc7a8*$^{-/-}$ F1 mice in the mixed C57BL6/J-129Sv strain for 10 generations with pure C57BL6/J wild-type mice alternating male and females to avoid a genetic drift in the X and Y chromosomes.

## Genotyping

Mice genotype was confirmed by triplex-PCR using DNA from the tail. Primers used were forward: 5'GGAGCGATCTGCGGAGTGA3'; reverse: 5'ACAGAGTGCGCTCCTACCCT3' and reverse KO-specific: 5'CGGTGGGCTCTATGGGTCTA3', and Standard DNA polymerase (*Biotools* Ref:10.002). The PCR products are 458 bp (wild type allele) and 180 bp (*Slc7a8*$^{-/-}$ allele) fragment.

## Protein analysis

Protein analysis was done by western blotting using total membrane samples. Frozen tissues (50–100 mg) were homogenized in 5 mL of membrane buffer (25 mM HEPES – 4 mM EDTA – 250 mM sucrose – and protease inhibitors) and centrifuged at 10,000 rpm for 10 min at 4°C. Supernatant was centrifuged at 200,000xg for 1 hr at 4°C. The pellet was resuspended in 150 µL of membrane buffer using a 25G syringe. Pierce BCA Protein Assay Kit (Thermo Scientific Ref:23225) was used for protein quantification. Polyclonal rabbit antibody against mouse SLC7A8 protein was generated using an antigen against the C-terminal region (peptide sequence: PIFKPTPVKDPDSEEQP) (*Figure 1—figure supplement 1B*). Serum extracts from inoculated rabbits were purified with protein G and used as primary antibody. Detection was by chemiluminescent reaction using ECL (GE Healthcare Ref: RPN2232) and autoradiography (Amersham Hyperfilm Ref:28906839). For specific SLC7A8 light subunit detection, samples were run in the presence of 100 mM of dithiothreitol (SigmaAldrich Ref: D9779).

## Behavior tests

**Rotarod** (Panlab Ref:LE8500). The experimental design consisted of two training trials (TR) at the minimum speed (4 rpm) followed of two different tasks: (a) motor coordination and balance were assessed by measuring the latency to fall off the rod in consecutive trials with increasing fixed rotational speeds (FRS 4, 10, 14, 19, 24, and 34 rpm). The animals were allowed to stay on the rod for a maximum period of 1 min per trial and a resting period of 5 min was left between trials. (b) In the accelerating rod test, the rotation speed was increased from 4 to 40 rpm during two sessions of 1 min. For each trial, the elapsed time until the mouse fell off the rod was recorded. **Treadmill** (Panlab Ref:LE8710MTS): During two training trials (TR), the inclination of the treadmill was increased from 0° to 20° from the horizontal plane at different speeds (5, 10, 20, 30, 40 and 50 cm/s). Whenever an animal fell off the belt, foot shocks were applied for a maximal duration of 1 s. After the shock, mice were retrieved and placed back. **Morris water maze** (MWM): Mice were tested over 4 days (four trials/session, 10 min inter-trial intervals). The Morris Water Maze test consists of a circular tank (150 cm diameter, 100 cm high) filled with opaque water (with non-toxic white paint) and maintained at 21 ± 2°C. A removable circular platform (8 cm diameter) was located in a fixed position (NE quadrant) inside the pool. The pool was surrounded by white curtains, with cues affixed. The test was performed under low non-aversive lighting conditions (50 lux). An overhead camera connected to video-tracking software (SMART, Panlab SL., Spain) will be used to monitor the animal's behavior. Latency to reach the platform, total distance travelled, speed and time in zones will be recorded for

posterior data analysis. The maze was surrounded by white curtains with black patterns affixed, to provide an arrangement of spatial cues. A pre-training session was performed in which the platform was visible in the center (day 1), followed by five acquisition sessions during which the platform was submerged 2 cm below the water (days 2–6). In each trial, mice were introduced in the pool from one of the random starting locations. Mice failing to find the platform within 60 s. were placed on it for 10 s. At the end of every trial the mice were dried for 15 min in a heater. Escape latencies, length of the swimming paths and swimming speed for each animal and trial were monitored and computed by a tracking system connected to a video camera placed above the pool. **Pre-pulse inhibition of acoustic startle response** (PPI) (Panlab Ref:LE116): Training was 5 min of habituation time to the apparatus with a background noise level of 70 dB and then exposed to six blocks of 7 trial types in pseudo-random order with 15 s. inter-trial intervals. The trials: 1 s of a 120 dB, 8000 kHz sound preceded 100 ms. by a 40 ms pre-pulse (PP) sound of 74, 78, 82, 86 or 90 dB. The startle response was recorded for 65 ms, measuring every 1 ms. from the onset of the startle stimulus. **Restrain stressor** (LabResearch Ref:G05): Mice were habituated for 3 days prior the experiment collecting 10–15 µL of blood from tail. All sets were carried in the same room at the same time to minimize environmental variations and corticosterone fluctuations as a result of circadian rhythms. Mice were placed for 15 min in the conditioning unit and 75 µL of tail's blood was collected. For recovery mice were placed into a clean cage for 90 min. Blood corticosterone were determined by Corticosterone EIA kit (Enzo Ref:ADI900097).

## Auditory brainstem response test (ABR)

Hearing was evaluated by recording the auditory brainstem responses (ABR) with a System 3 TDT Evoked Potential Workstation (Tucker Davis Technologies TDT, Alachua, FL, USA) as previously described (*Cediel et al., 2006*; *Riquelme et al., 2010*). Briefly, mice were anesthetized with intraperitoneal injection of ketamine (100 mg/kg) and xylazine (10 mg/kg), and placed inside a sound chamber. Broadband click (0.1 ms) and tone bursts (5 ms) at 8, 16, 20, 28 and 40 kHz were delivered with an open field speaker (MF1, TDT) at an intensity range from 90 to 10 dB sound pressure level (SPL) in 5–10 dB SPL steps. The electrical responses were amplified and averaged and the ANABR recordings analyzed with BioSig software (TDT) to determine hearing thresholds in response to each stimulus, peak and interpeak latencies and peak amplitudes. Animals were kept thermostatized and monitored during both anesthesia and the following recovery period.

## Histology and immunohistochemistry

Mice were perfused through vascular system with 4% PFA and inner ear and brain samples were collected. The cochlea was dissected, post-fixed and decalcified in 0.3 M EDTA pH 6.5 (Sigma-Aldrich Ref:E1644) for seven days. Decalcified cochleae were embedded in OCT or paraffin as reported (*Murillo-Cuesta et al., 2012*). Deparaffinized cochlear sections were stained with hematoxylin and eosin for general cytoarchitecture evaluation. **Immunohistochemistry**: Floating brain tissue sections were incubated with 3% $H_2O_2$ in 10% methanol in PBS for 10 min. Blocking buffer with: 0.2% gelatine, 0.2% Triton x-100% and 10% FBS for 30 min. Primary antibody: anti-SLC7A8 1/500 in blocking buffer ON at 4°C with agitation. Secondary antibody: 1/200 biotinylated anti rabbit in blocking buffer for 1 hr at RT. Third antibody: 1/100 of A + B conjugate (Vectastain, Ref:ABC kit) in blocking buffer for 1 hr at RT. Develop staining: 0.03%DAB in PBS for 5 min. Reaction: incubate 0.03% DAB +1/10.000 $H_2O_2$ for 2–7 min with agitation. Reaction was stopped by rinsing with PBS. Sections were dried and dehydrated before mounting. Detection was using a bright-light microscope. **Immunofluorescence:** OCT tissue sections were permeabilized by incubating for 10 min with 0.1% Triton X-100 and incubated as reported (*Sanchez-Calderon et al., 2010*; *de Iriarte Rodríguez et al., 2015b*) with the following primary antibodies: anti-SLC7A8 (1/200), -s100 (1/1000, Sigma-Aldrich Ref:S2532), -Kir4.1 (1/200, Merck Millipore Ref:AB5818), -IBA1 (1/100, Abcam Ref:ab5076), or with Phalloidin (1/100, Thermo Fisher Scientific Ref:A22287), ON at 4°C. Sections were then incubated with secondary antibodies: (1:300, Thermo Fisher Scientific Ref:A-11034 Goat anti-Rabbit Alexa Fluor 488, Ref:A-11030 Goat anti-Mouse Alexa Fluor 546, Ref:A-21206 Donkey anti-Rabbit Alexa Fluor 488, Ref:A-11056 Donkey anti-Goat Alexa Fluor 546) for 2 hr at RT. Detection by confocal microscopy (Leica, Ref:LSM 780 Zeiss).

## Fluorescence quantification

Four sections of apex, middle and basal turns of the cochlea were quantified using the same settings, including argon laser voltage, for the quantification. Using Fiji software, the sum of the intensity of all stacks (2.6 μm in the z axis along the 10 μm section) from the spiral ligaments + stria vascularis area was extracted. Data were analyzed with Prism 7 statistic software package (Graph Pad Software, Inc.). Statistical significance was determined by Student's t test for unpaired samples. The number of biological and experimental replicates are detailed in the legend of each figure.

## Quantitative RT-PCR

RNA was isolated using RNeasy (Qiagen) from 1 to 2 cochleae; its integrity and concentration were assessed using an Agilent Bioanalyzer 2100 (Agilent Technologies). At least, three mice per condition were used. cDNA was then generated by reverse transcription (High Capacity cDNA Reverse Transcription Kit; Applied Biosystems) and gene expression analyzed in triplicate by qPCR using TaqMan Gene Expression Assay kits (Applied Biosystems). The following probes were used: potassium voltage-gated channel subfamily Q member 2 (*Kcnq2*) Mm00440080_m1; potassium voltage-gated channel subfamily Q member 3 (*Kcnq3*) Mm00548884_m1; potassium voltage-gated channel subfamily Q member 5 (*Kcnq5*) Mm01226041_m1; prestin (*Slc26a5*) Mm00446145_m1; T-box transcription factor TBX18 (*Tbx18*) Mm00470177_m1; interleukin 1 beta (*Il1b*) Mm00434228m1; interleukin 6 (*Il6*) Mm00446190m1; solute carrier family 7 (cationic amino acid transporter, y + system), member 8 (*Slc7a8*) Mm01318971m1. PCR was performed on an Applied Biosystems 7900HT Real-Time PCR System using *Hprt1 or RPLP0* as the endogenous housekeeping gene. Relative quantification values were calculated using the 2-ΔΔCt method. All procedures have been already reported (*de Iriarte Rodríguez et al., 2015a*).

## ARHL cohort recruitment and clinical assessment

A total of 147 Subjects were recruited in North-Eastern Italy isolated villages (FVG Genetic Park) (*Esko et al., 2013*) and from one isolated village from Southern Italy (Carlantino). Subjects underwent a clinical evaluation to exclude any syndromic form of hearing loss or other systemic illnesses linked with sensorineural hearing loss. Audiometric tests using standard audiometers were carried out for each subject. Measurements have been obtained after any acoustically obstructing wax was removed. Thresholds for six different frequencies (0.25, 0.5, 1, 2, 4, 8 kHz) were measured and then a pure-tone average for high frequencies (P-TAH) was computed by taking the average of 4 and 8 kHz. To avoid non-genetic variations in the hearing phenotype (e.g. monolateral hearing loss), the best hearing ear was considered for each individual. Cases were defined as people older than 50 years old having PTAH ≥40, while controls were subjects more than 50 years old with PTAH ≤25.

All studies were approved by the Institutional Review Board of IRCCS Burlo Garofolo, Trieste, Italy and consent forms for clinical and genetic studies have been signed by each participant. All research was conducted according to the ethical standards as defined by the Helsinki Declaration.

## Whole genome sequencing and mutation screening

Blood samples were collected and used to extract DNA using standard protocols. Low coverage whole genome sequence was generated using Illumina technology (Genome Analyzer and HiSeq 2000) at the Welcome Trust Sanger Institute and Beijing Genomics Institute. Data coverage was ranging from 4 to 10X. A multi-sample genotype calling was performed and standard quality filters were applied. The detailed pipeline has already been described elsewhere (*Timpson et al., 2014*). Variants belonging to *SLC7A8* gene were extracted using bcftools [http://samtools.github.io/bcftools/] and annotated with ANNOVAR (*Wang et al., 2010*). Only the exonic variants were further considered. Finally, variants of interest were confirmed by direct Sanger sequencing on a 3500 Dx Genetic Analyzer (Life Technologies, CA), using ABI PRISM 3.1 Big Dye terminator chemistry (Life Technologies) per manufacturer's instructions. Mutation frequencies were compared with public databases such as Esp6500siv2 (NHLBI Exome Sequencing Project), 1000 g (1000 Genomes Project), Campion (The Allele Frequency Net Database) and ExAC (The Exome Aggregation Consortium). For SLC7A8 we collected several statistics including the probability of loss of function intolerance (pLI), where the closer pLI value is to 1, the more LoF intolerant the gene could be considered. We also

collected the missense Z score, a positive score indicates intolerance to missense variation whereas a negative Z score indicates that the gene had more missense variants than expected.

## Site-directed mutagenesis

The QuikChange site-directed mutagenesis kit (Stratagene) was used to introduce point mutations in SLC7A8 sequence, according to the manufacturer's protocol. The pcDNA3.1-StrepTag fused SLC7A8 construct was used as template (Costa et al., 2013). Amino acid substitutions were introduced into SLC7A8 sequence using a compatible reverse primer and forward primers (Figure 5—source data 1). All primers annealed to the coding sequence, and the position of the mutated codon was underlined. All constructs were verified by DNA sequencing and then used for transient transfection.

## Cell culture and transfection

HeLa cells (Sigma Aldrich, Ref: 93021013) were maintained at 37°C/5% CO2 in Dulbecco's modified Eagle's medium (Life Technologies) supplemented with 10% (v/v) fetal bovine serum, 50 units/ml penicillin, 50 µg/ml streptomycin, and 2 mM l-glutamine. HeLa cells were transiently transfected with plasmid constructions mentioned above with the use of Lipofectamine 2000 (Invitrogen) following the manufacturer's protocol. Amino acid transport and fluorescence microscopy analyses were carried out 48 hr after transfection.

## Visualization of Strep-tagged amino acid transporters by fluorescence microscopy

To analyze the effect of the mutations on SLC7A8 protein expression and plasma membrane localization, fluorescence microscopy of Strep-tagged wild type and mutant transporters was performed on a semiconfluent monolayer of transfected HeLa cells cultured on glass coverslips. Glass coverslip-grown cells were incubated with 1 mg/ml wheat germ agglutinin (WGA) labeled with Texas-Red (Thermo Fisher Scientific) at 37°C for 10 min, rinsed three times with phosphate-buffered saline-$Ca^{2+}$-$Mg^{2+}$ and fixed for 15 min in 4% paraformaldehyde. Fixed cells were blocked in blocking buffer (10% FBS and 0.1% saponin in PBS) for 1 hr and then incubated for 1 hr with primary antibody (anti-Strep Tag GT517, 1/100; Abcam). Secondary goat-anti-mouse-FITC antibody (Life Technologies) was incubated for 2 hr protected from light and rinsed three times with phosphate-buffered saline. Nuclear staining was performed by incubating 1 µg/ml Hoechst (Thermo Fisher Scientific) for 10 min, rinsed three times with phosphate-buffered saline and then mounted with aqua-poly/mount coverslipping medium (Polysciences Inc.). Images were taken using a Nikon E1000 upright epifluorescence microscope. All images were captured during 200 ms except for those corresponding to V460E that were overexposed to 2 s to reveal the subcellular localization of this very low expressing variant. To quantify SLC7A8 wild type and mutated transporters expression levels in cells, a single in-focus plane was acquired. Using ImageJ (v1.48, NIH), an outline was drawn around each cell and area and mean fluorescence measured, along with several adjacent background readings. The total corrected cellular fluorescence (TCCF) = integrated density – (area of selected cell × mean fluorescence of background readings), was calculated.

## Amino acid transport assay

Amino acid uptake was measured by exposing replicate cultures at room temperature to L- [³H]-labeled alanine or [³H]-tyrosine (1 µCi/ml; Perkin Elmer) in sodium-free transport buffer (137 mM choline chloride, 5 mM KCl, 2 mM $CaCl_2$, 1 mM $MgSO_4$, and 10 mM HEPES, pH 7.4). Initial rates of transport were determined using an incubation period of 1 min and 50 µM of cold alanine or tyrosine. Assays were terminated by washing with an excess volume of chilled transport buffer. Cells were lysed using 0.1% SDS and 100 mM NaOH and radioactivity measured in a scintillation counter. Uptake values were corrected by their total corrected cellular fluorescence (TCCF) for all transporters except for V460E mutant, which does not reach the plasmatic membrane.

## Statistical analysis

Behavior and ABR experiments using mice were not performed blind to genotype and treatment conditions, but as data acquisition was automated this will not affect data processing and analysis.

The sample size was chosen according to the standard sample sizes used in the field and without applying any statistical method. The general criteria of exclusion were pre-established: (1) samples with a value that differed by more/less than two standard deviations from the mean value were excluded from the study. The statistical tests used in each experiment were appropriate to the type of groups, data and samples. Unpaired Student t-test was used for experiments with only two independent groups. Repeated measures two-way ANOVA was applied when we had to compare two independent groups (genotype as the between subjects factor) where repeated measurements of the dependent variable were obtained (Rotarod and PPI).

Data were analyzed with IBM SPSS 23.0 statistic software package (Chicago, IL). Statistical significance was determined by one-way analysis of variance (ANOVA) and Levene's F test to assess the equality of variances. When significant differences were obtained, post hoc comparisons were performed using Bonferroni or Tamhane tests to compare the three genotypes. Normal distribution of data and homogeneity of variances was assessed using Shapiro-Wilk and Levene tests, respectively. In most of the datasets these two assumptions were achieved. However, when not achieved and because we use comparable sample sizes and ANOVA is robust to normality violations, our results are still valid. Sphericity assumption was assessed using Mauchly's test and when not achieved Greenhouse correction was taken. Posthoc tests were performed using Bonferroni correction for individual comparisons. Bonferroni $p < 0.05$ was assumed as critical value for significance throughout the study. Statistical analyses were performed using SPSS package.

## Acknowledgements

We thank to all volunteers that participate in this study, Professor Fernando Aguado for technical assistance in brain IHC, Professor Josep Chillarón for SLC7A8 antibody design, Dr. Miguel López de Heredia for critical reading of the manuscript, Saif Ahmed for the code to extract the images' analysis data, the core-facilities: IRB Advanced Digital Microscopy Facility (Anna Lladó and Julien Colombelli), the Genomics facility at the IIBm (CSIC-UAM), Units F1, F3 and F6-SEFALer (CIBERER) for their technical support and GenOway for the help in the homologous recombination and transfection of the construct into ES cells to generate the $Slc7a8^{-/-}$ mice. IRB Barcelona is the recipient of a Severo Ochoa Award of Excellence from MINECO (Government of Spain). MD acknowledges support of the Spanish Ministry of Economy and Competitiveness, 'Centro de Excelencia Severo Ochoa 2013–2017; MD: and V N acknowledge the support of the CERCA Programme/Generalitat de Catalunya.' Isabel Varela-Nieto, Paolo Gasparini, Manuel Palacín and Virginia Nunes share leadership.

## Additional information

### Funding

| Funder | Grant reference number | Author |
| --- | --- | --- |
| Qatar National Research Fund | JSREP07-013-3-006 | Meritxell Espino Guarch |
| CIBERER | ACCI 2016 | Silvia Murillo-Cuesta<br>Manuel Palacín<br>Virginia Nunes |
| CIBERER | ACCI 2017 | Silvia Murillo-Cuesta<br>Manuel Palacín<br>Virginia Nunes |
| Generalitat de Catalunya | SGR 2014/1125 | Mara Dierssen |
| Ministerio de Ciencia e Innovación | SAF2016-79956-R | Mara Dierssen |
| Seventh Framework Programme | TARGEAR FP7-PEOPLE-2013-IAPP | Isabel Varela-Nieto |
| Ministerio de Ciencia e Innovación | SAF2014-53979-R-FEDER | Isabel Varela-Nieto |
| Ministerio de Ciencia e Innovación | SAF2015-64869-R-FEDER | Manuel Palacín |

| Generalitat de Catalunya | SGR2009-1355 | Manuel Palacín |
| Spanish Health Institute Carlos III-FIS | PI13/00121-R-FEDER | Virginia Nunes |
| Generalitat de Catalunya | SGR2009-1490 | Virginia Nunes |
| Spanish Health Institute Carlos III-FIS | PI16/00267-R-Feder | Virginia Nunes |

The funders had no role in study design, data collection and interpretation, or the decision to submit the work for publication.

### Author contributions
Meritxell Espino Guarch, Conceptualization, Resources, Methodology, Writing—original draft, Writing—review and editing, Western Blot, Brain IHC, Stress Response, IF and H/E quantifications; Mariona Font-Llitjós, Conceptualization, Methodology, Writing—review and editing, Lat2-/- generation Behavior tests; Silvia Murillo-Cuesta, Formal analysis, Writing—review and editing, ABR tests Cochlear RT-qPCR; Ekaitz Errasti- Murugarren, Methodology, Writing—review and editing, In vitro uptake; Adelaida M Celaya, Methodology, Writing—review and editing, Cochlear sampling and cytoarchitecture Cochlear H/E and IF Cochlear RT-qPCR; Giorgia Girotto, Methodology, Writing—review and editing, Subject samples; Dragana Vuckovic, Massimo Mezzavilla, Formal analysis, Writing—review and editing, Mutations screening; Clara Vilches, Methodology, Writing—review and editing, Stress Response; Susanna Bodoy, Methodology, Writing—review and editing, Western Blot; Ignasi Sahún, Methodology, Writing—review and editing, Behavior tests; Laura González, Methodology, Mice maintenance and necropsy; Esther Prat, Methodology, Cochlear H/E and IF Mice maintenance and necropsy; Antonio Zorzano, Conceptualization, Writing—review and editing; Mara Dierssen, Conceptualization, Resources, Methodology, Writing—review and editing, Behavior tests; Isabel Varela-Nieto, Conceptualization, Resources, Writing—review and editing; Paolo Gasparini, Conceptualization, Resources, Writing—review and editing, Subject samples; Manuel Palacín, Conceptualization, Resources, Supervision, Writing—original draft, Writing—review and editing; Virginia Nunes, Conceptualization, Resources, Funding acquisition, Writing—original draft, Writing—review and editing

### Author ORCIDs
Meritxell Espino Guarch (iD) https://orcid.org/0000-0002-7211-3229
Silvia Murillo-Cuesta (iD) http://orcid.org/0000-0002-8706-4327
Mara Dierssen (iD) http://orcid.org/0000-0003-0853-6865
Virginia Nunes (iD) http://orcid.org/0000-0002-5747-9310

### Ethics
Human subjects: Ethical Consent Published in: Esko T, Mezzavilla M, Nelis M, Borel C, Debniak T, Jakkula E, et al. Genetic characterization of northeastern Italian population isolates in the context of broader European genetic diversity. Eur J Hum Genet. 2013;21(6):659-65.
Animal experimentation: Animal experimentation complied with the ARRIVE guidelines and was conducted in accordance with Spanish (RD 53/2013) and European (Directive 2010/63/EU) legislations. All protocols used in this study were reviewed and approved by the Institutional Animal Care and Use Committee at IDIBELL in a facility accredited by the Association for the Assessment and Accreditation of Laboratory Animal Care International (AAELAC accredited facility, B900010). Mice procedures were done according with scientific, humane, and ethical principles. The studied mouse model did not show phenotype differences comparing male and female. Thus, to ensure that our research represents both genders, the studies describes in this work were performed using both sexes equitably. The number of biological and experimental replicates is detailed in the legend of each figure.

### Decision letter and Author response
Decision letter https://doi.org/10.7554/eLife.31511.026
Author response https://doi.org/10.7554/eLife.31511.027

## Additional files

### Supplementary files
• Transparent reporting form
DOI: https://doi.org/10.7554/eLife.31511.021

### Major datasets
The following previously published dataset was used:

| Author(s) | Year | Dataset title | Dataset URL | Database, license, and accessibility information |
|---|---|---|---|---|
| Esko T, Mezzavilla M, Nelis M, Borel C, Debniak T, Jakkula E, Julia A, Karachanak S, Khrunin A, Kisfali P, Krulisova V, Ausrele Kucinskiene Z, Rehnstrom K, Traglia M, Nikitina-Zake L, Zimprich F, Antonarakis SE, Estivill X, Glavac D, Gut I, Klovins J, Krawczak M, Kucinskas V, Lathrop M, Macek M, Marsal S, Meitinger T, Melegh B, Limborska S, Lubinski J, Paolotie A, Schreiber S, Toncheva D, Toniolo D, Wichmann HE, Zimprich A, Metspalu M, Gasparini P, Metspalu A | 2013 | Genetic characterization of northeastern Italian population isolates in the context of broader European genetic diversity | https://www.ebi.ac.uk/biostudies/studies/S-EPMC3658181 | Publicly available at EBI BioStudies (accession no: S-EPMC3658181) |

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
