## [Decision Letter]

Thank you for submitting your manuscript "Mutations in LAT2 amino acid transporter support SLC7A8 as a novel gene involved in Age-Related Hearing Loss" to *eLife*. The manuscript has been reviewed by three expert reviewers, and their assessments together with my own, forms the basis of this letter. I am also including the three reviews in their original form at the end of this letter, as there are many specific and useful suggestions in them that will not be repeated in the summary here.

We would like to encourage you to resubmit a revised manuscript that addresses the specific issues raised in the reviews. We appreciate that the reviewers' comments cover a broad range of suggestions for improving the manuscript. As you know from our earlier correspondence, the nomenclature confusion needs to be clearly addressed at the start of the manuscript. Please use the proper nomenclature for the mouse and human genes: Slc7a8 and SLC7A8, respectively. You can mention the history of the nomenclature confusion in one sentence and then delete LAT2/Lat2 in the title and in all other locations in the manuscript. The HUGO Gene Nomenclature Committee allows for only one human official gene name and that is SLC7A8 (italicized). See https://www.genenames.org/cgi-bin/search?search_type=all&search=Slc7a8&submit=Submit.

In addition to the various specific comments, there was a general consensus among the reviewers that a more complete assessment of the human phenotype would substantially strengthen this work. Also, correlating the inner ear histology of mice with phenotypes of varying severity would strengthen this work.

Reviewer #1

This is a well-documented study showing that loss of the amino acid transporter Lat2 produces an age-dependent hearing deficit. The knockout clearly impairs hearing as documented by behavioral and clinical electrophysiological tests. Surprisingly, the penetrance is far from complete, but the authors extend the analysis to an inbred strain that already exhibits some age-dependent hearing loss, and show that the penetrance increases. However, it is also clear that the loss of Lat2 has little additional effect on hearing at the earlier time point (4-6 months) and no effect at the later (7-13 months) (Figure S4C,D). The text describes this somewhat misleadingly as increased penetrance, but this seems only due to the underlying defect in this inbred strain, not any increased effect of the Lat2 KO. The reduced penetrance is of concern, and it is good that the authors attempt to address this, but the use of this inbred strain does not seem to provide much relevant information, other than that the two manipulations (Lat2 KO and strain) seem to involve the same process since there is no additive effect.

The analysis of sequence variation in affected individuals and the control population provides strong evidence that the changes in Lat2 are pathogenic. The effects on transport activity are clear and interesting, although a little difficult to correlate with the structure.

The only area in which the work falls short is the understanding of mechanism. The authors do document the effects of Lat2 loss on cochlear morphology, and the analysis of gene expression as well as histology is thoughtful and convincing. However, it still remains unclear how mutations in Lat2 might lead to this phenotype. The authors consider a number of reasonable possibilities, but hair cell recordings, even from younger animals, might be extremely informative. Even without knowing the precise mechanism, and even though it remains unclear where the defect arises, the hair cells appear severely affected and any evidence for a physiological effect in the mice would greatly increase the impact of this work.

Reviewer #2

In the process of reading and critiquing the manuscript (Gaurch et al., 2017), I first provided comments below about the phenotyping of the Lat2 mutant mouse and made some suggestions (numbered 1-11) for improving the manuscript. I then read the Results and Discussions section about variants of human "LAT2/SLC7A8". Here, there is a mistake that seems so basic I doubted myself. In the Discussion and in the heading of Table 1, as examples, the authors assume that LAT2 and SLC7A8 are the same gene. In the first paragraph of the Discussion section, there are the following three sentences. "Here, we show that loss-of-function of the amino acid transporter LAT2 is associated with ARHL in both humans and mice..………….. In patients with ARHL we identified four LAT2 variants that showed loss-of-function of transport of tyrosine (Figure 5). Altogether, these results indicate that full LAT2 function is needed to keep an optimal hearing function throughout life, with half a dose of LAT2 being enough to accelerate ARHL phenotype in mice and humans." Also, consider the title of Table 1, which is "LAT2 (SLC7A8) humans mutations found in ARHL and controls individuals" [note typos]. The authors make the explicit assumption that LAT2 and SLC7A8 are one in the same gene. Additionally, Figure 5 is titled "in vitro characterization of LAT2 mutants". The authors listed some of the variants of LAT2 such as V302I, T402M, R418, and V460E despite the fact that the longest isoform of human LAT2 only encodes 243 residues (http://genome.ucsc.edu/cgi-bin/hgGene?hgg_gene=uc003uai.4&hgg_prot=ENST00000344995.9&hgg_chrom=chr7&hgg_start=74209395&hgg_end=74229834&hgg_type=knownGene&db=hg38&hgsid=610631709_QOLacwy0rMjb5LhLF9wFK9GjFcYA).

As one reads the entire manuscript, the authors assume incorrectly that LAT2/Lat2 and SLC7A8/Slc7a8 are the same gene. But, that can't be true both for mouse and for human. The mouse Lat2 gene is on chromosome 5 and encodes a protein of 203 residues, while the mouse Slc7a8 is located on chromosome 14. The human LAT2 gene is located on chromosome 7. The human SLC7A8 gene encodes a protein of 535 amino acid residues and resides on chromosome 14. Clearly different genes. What this means is that the knockout mouse model of Lat2 and its phenotype does not inform us about the putative pathogenicity of the heterozygous variants of human SLC7A8. The authors have merged two distinct stories as a result of a very basic error.

All of the comments below were written before this reviewer noticed the above mentioned fatal flaw.

1) In Figure 1 add a simple schematic diagraming the nature of the Lat2 mutation and how the variant results in a "knockout". Reference 26 is in Spanish. The reviewer understands that in the lanes with protein from a homozygous knockout mouse, no LAT2 protein is detected. However, there is no indication in the legend where the epitope is located for the antibody that was used against LAT2. In subsection “Genotyping”, the authors state that a peptide was used to generate polyclonal antisera and the peptide is located near the C-terminus. This is important information needs to be provided in a simple diagram in Figure 1 of the Lat2 gene and LAT2 protein so that data can be easily interpreted without hunting around.

2) In the Materials and methods section, the authors state that exon 1 of Lat2 was replaced with a neomycin cassette. Is there any evidence for an alternative translation start codon downstream of exon 1 that is used in the absence of exon 1. As shown in the UCSC Browser (http://genome.ucsc.edu/cgi-bin/hgTracks?db=mm10&lastVirtModeType=default&lastVirtModeExtraState=&virtModeType=default&virtMode=0&nonVirtPosition=&position=chr5%3A134596883%2D134619265&hgsid=610482073_vc2WSLDW4MAetOvl4BmAD4fijN8N), Lat2 has two different first exons. One exon 1 is entirely noncoding and the other exon 1 includes a 5' UTR and some protein coding sequence. Which exon 1 was deleted? Provide an accession number for your gene structure of Lat2 and a diagram in Figure 1 so that it is clear what was deleted in your mutant mouse.

3) Subsection “Localization and quantification of LAT2 in the inner ear”. Provide a reference for "the previous study" and state where the previous study showed localization of LAT2 that you think may be mislocalization. I know you are being courteous here but more information would be helpful. Also considering that there are at least two alternative splice protein coding isoforms of LAT2 (with and without exon 6 of 12), might there be another explanation for the discrepancy? Where was the epitope of LAT2 located in the other study?

4) Results section, the authors state that there is a significant reduction in latency in the rotorod acceleration test. Latency of what? Similarly (Results section), explain why an "increased exposure to shock on the treadmill" is important and what it means biologically. Just stating it represents "motor coordination performance" and PPI may be useful information for those in the field but to others it is jargon/lingo.

5) Figure 1, panel A, there are no loading controls in the lanes of the Western blot. In the legend it is noted that 50ug of protein was loaded in each lane. That is helpful information but insufficient.

6) Figure 1, panel B and Results section, the authors need to explain what is "Pre-Pulse inhibition of the acoustic startle response (PPI)" and what is being measured.

7) Figure 1, panels C and F, on the Y-axis, indicate that click thresholds are ABR thresholds.

8) Figure 1, The Y-axis is without a metric/description.

9) Figure 2 also needs a diagram to help orient a reader to the structures and cell types of the inner ear. Were the immunofluorescent images acquired and evaluated with the investigator blinded to genotype? Panel B is nice but not sufficient. In panel C, add a key in the figure panel for the three bar types.

10) Figure 2, panel D, and subsection “Localization and quantification of LAT2 in the inner ear”, which states that there is a progressive increase in LAT2 expression. After P1, the up and down levels of expression seem more like experimental noise and not a progressive increase. How many independent determinations of 3 mice per group of pooled samples were evaluated? Are the error bars for technical replicates?

11) Figure 5, legend, it is not clear what is meant by "[…]unless V60E that[…]" and "[…]is represented the uptake[…]"

Reviewer #3

This is an interesting and novel paper describing the pathogenicity of a novel gene LAT2 in age-related hearing loss in both mice and humans. The authors first examined LAT2-deficient mice and found an incomplete penetrance of the homozygous animals showing age-related threshold shifts, although these results were obtained in mouse strains that suffer from age-related hearing loss (C57Bl6) rather than CBA or FVB strains. Interestingly some heterozygous animals also demonstrate elevated hearing thresholds thus suggesting a dosage effect. LAT2 expression was found in several locations within the cochlea including the stria vascularis, spiral ligament and SGN. Cell loss concomitant with decreased in specific genes in LAT2+ areas were found. Mechanistically, in vitro data in HeLA cells indicate that amino acid transport is disrupted in mutated LAT2, which agrees with a more generalized metabolic derangement in the cochlea though this was not demonstrated in vivo. Next, the authors identified a cohort of patients with age-related hearing loss and found that mutations leading to amino acid transport dysfunction in vitro were indeed ones associated with age-related hearing loss in patients.

Overall, the manuscript is well written, contains potentially exciting yet somewhat incomplete dataset. I recommend a major revision.

1) Hearing tests in families of patients with LAT2 mutations. It seems prudent to test both the genetics and hearing of family members to determine the genotype-phenotype relationship to ascertain if LAT2 mutations are necessarily pathogenic, and a dosage effect can also be supportive of such a conclusion. Also, LAT2 qPCR expression should be assessed in heterozygotes if a dosage effect is proposed.

2) Does histology correlate with hearing loss in mice? The incomplete penetrance and use of C57bl6 to examine an age-related hearing loss candidate gene is problematic. But a correlation between the degree of cell loss and hearing loss can be supportive evidence yet this is lacking. Also for some of the cell markers, assessing the intensity of immunostaining seems odd. One should instead quantify the number of immunostained cells. Lastly, the qPCR results of several genes on the LAT2 were missing.

---

## [Author Response]

Reviewer #1This is a well-documented study showing that loss of the amino acid transporter Lat2 produces an age-dependent hearing deficit. The knockout clearly impairs hearing as documented by behavioral and clinical electrophysiological tests. Surprisingly, the penetrance is far from complete, but the authors extend the analysis to an inbred strain that already exhibits some age-dependent hearing loss, and show that the penetrance increases. However, it is also clear that the loss of Lat2 has little additional effect on hearing at the earlier time point (4-6 months) and no effect at the later (7-13 months) (Figure S4C,D). The text describes this somewhat misleadingly as increased penetrance, but this seems only due to the underlying defect in this inbred strain, not any increased effect of the Lat2 KO. The reduced penetrance is of concern, and it is good that the authors attempt to address this, but the use of this inbred strain does not seem to provide much relevant information, other than that the two manipulations (Lat2 KO and strain) seem to involve the same process since there is no additive effect.

We have now finalized the longitudinal study of the onset of hearing loss up to 5 months of age in the inbred strain C57BL6/J (new Figure 2—figure supplement 2). At early age (<2.5 and 3.5 month-old) homozygous *Slc7a8*^-/-^ mice showed increased ABR thresholds and hearing loss penetrance when compared with either wild type or heterozygous *Slc7a8*^+/-^ mice. These differences are less marked at the age of 5 months. In this pure genetic background, heterozygous *Slc7a8*^+/-^ and wild type mice showed no evident no differences in ABR thresholds and hearing loss penetrance. In one hand, these new data confirm data presented in Figure 1—figure supplement 5 with different groups of mice studied at different ages and shows clearly that ablation of LAT2 anticipates the onset of hearing loss in C57Bl6/J mice. Moreover, *Lat2* knockout, *Slc7a8*^-/-^, mice showed higher penetrance but not more severity on the hearing loss phenotype when compared with the mixed genetic background strain (Figure 1).

Text was corrected (subsection “Slc7a8 ablation causes ARHL”) to: Additionally, longitudinal study of Slc7a8^-/-^ mice into the inbred C57BL6/J genetic background showed higher penetrance than the mixed background throughout the ages studied (Figure 2—figure supplement 2).

The analysis of sequence variation in affected individuals and the control population provides strong evidence that the changes in Lat2 are pathogenic. The effects on transport activity are clear and interesting, although a little difficult to correlate with the structure.

The only area in which the work falls short is the understanding of mechanism. The authors do document the effects of Lat2 loss on cochlear morphology, and the analysis of gene expression as well as histology is thoughtful and convincing. However, it still remains unclear how mutations in Lat2 might lead to this phenotype. The authors consider a number of reasonable possibilities, but hair cell recordings, even from younger animals, might be extremely informative. Even without knowing the precise mechanism, and even though it remains unclear where the defect arises, the hair cells appear severely affected and any evidence for a physiological effect in the mice would greatly increase the impact of this work.

The reviewer suggestion is right; indeed, these measurements will complement the data shown here. Hair cell recording is a highly specialized and challenging technique that is not set up in our labs. However, we have considered the reviewers’ point and now the correlation between hearing loss phenotype and cochlear damage is shown (Figure 3—figure supplement 1).

Text was corrected (subsection “Lack of *Slc7a8* induced damage in the organ of Corti, spiral ganglion and stria vascularis”) to: Slc7a8^-/-^ mice 4- to 7-month old presented ~50% of cell loss in the spiral ganglion compared with wild type mice (Figure 3). Decreased number of cells in the ganglia significantly correlated with ABR threshold and HL phenotype (Figure 3—figure supplement 1).

Text was corrected (subsection “Lack of *Slc7a8* induced damage in the organ of Corti, spiral ganglion and stria vascularis”) to: Reinforcing this observation, the expression of Kir4.1, a potassium channel highly expressed in stria vascularis cells (33), was also dramatically reduced by 50% in Slc7a8^-/-^ mice (Figure 4 and Figure 4—figure supplement 1). Likewise, decreased expression of Kir4.1 marker correlates with HL phenotype (Figure 3—figure supplement 1).

Text was corrected (subsection “Lack of *Slc7a8* induced damage in the organ of Corti, spiral ganglion and stria vascularis”) to: Moreover, mice with severe HL phenotype showed 30% less number of fibrocytes in the spiral ligament (Figure 3—figure supplement 1).

Reviewer #2In the process of reading and critiquing the manuscript (Gaurch et al., 2017), I first provided comments below about the phenotyping of the Lat2 mutant mouse and made some suggestions (numbered 1-11) for improving the manuscript. I then read the Results and Discussions section about variants of human "LAT2/SLC7A8". Here, there is a mistake that seems so basic I doubted myself. In the Discussion and in the heading of Table 1, as examples, the authors assume that LAT2 and SLC7A8 are the same gene. In the first paragraph of the Discussion section, there are the following three sentences. "Here, we show that loss-of-function of the amino acid transporter LAT2 is associated with ARHL in both humans and mice..………….. In patients with ARHL we identified four LAT2 variants that showed loss-of-function of transport of tyrosine (Figure 5). Altogether, these results indicate that full LAT2 function is needed to keep an optimal hearing function throughout life, with half a dose of LAT2 being enough to accelerate ARHL phenotype in mice and humans." Also, consider the title of Table 1, which is "LAT2 (SLC7A8) humans mutations found in ARHL and controls individuals" [note typos]. The authors make the explicit assumption that LAT2 and SLC7A8 are one in the same gene. Additionally, Figure 5 is titled "in vitro characterization of LAT2 mutants". The authors listed some of the variants of LAT2 such as V302I, T402M, R418, and V460E despite the fact that the longest isoform of human LAT2 only encodes 243 residues (http://genome.ucsc.edu/cgi-bin/hgGene?hgg_gene=uc003uai.4&hgg_prot=ENST00000344995.9&hgg_chrom=chr7&hgg_start=74209395&hgg_end=74229834&hgg_type=knownGene&db=hg38&hgsid=610631709_QOLacwy0rMjb5LhLF9wFK9GjFcYA).As one reads the entire manuscript, the authors assume incorrectly that LAT2/Lat2 and SLC7A8/Slc7a8 are the same gene. But, that can't be true both for mouse and for human. The mouse Lat2 gene is on chromosome 5 and encodes a protein of 203 residues, while the mouse Slc7a8 is located on chromosome 14. The human LAT2 gene is located on chromosome 7. The human SLC7A8 gene encodes a protein of 535 amino acid residues and resides on chromosome 14. Clearly different genes. What this means is that the knockout mouse model of Lat2 and its phenotype does not inform us about the putative pathogenicity of the heterozygous variants of human SLC7A8. The authors have merged two distinct stories as a result of a very basic error.All of the comments below were written before this reviewer noticed the above mentioned fatal flaw.

In our study, we certainly knocked out the *Slc7a8* gene to generate the mouse model studied, which is the amino acid transporter SLC7A8 in both mouse and humans. LAT2 is the standard abbreviation for 2 genes, the one coding for L-type Amino Acid Transporter-2 or SLC7A8and the Linker for Activation of T-Cells Family Member 2 (see table below), a fact that has possibly mislead this reviewer.

As we mentioned previously, to avoid potential confusions to the readers, the SLC7A8 nomenclature is now used throughout the manuscript, and we clarify this point in the Introduction and include the gene reference in the key resources Table 2.

We have also modified the title of the manuscript to: Mutations in L-type amino acid transporter-2 support SLC7A8 as a novel gene involved in Age-Related Hearing Loss.

1) In Figure 1 add a simple schematic diagraming the nature of the Lat2 mutation and how the variant results in a "knockout". Reference 26 is in Spanish. The reviewer understands that in the lanes with protein from a homozygous knockout mouse, no LAT2 protein is detected. However, there is no indication in the legend where the epitope is located for the antibody that was used against LAT2. In subsection “Genotyping”, the authors state that a peptide was used to generate polyclonal antisera and the peptide is located near the C-terminus. This is important information needs to be provided in a simple diagram in Figure 1 of the Lat2 gene and LAT2 protein so that data can be easily interpreted without hunting around.

We have included the knockout strategy diagram and the information of the antibody epitope location (Figure 1—figure supplement 1) to further facilitate the non-Spanish readers the information described in the Published Thesis referenced in (Science AIfB. Allen Mouse Brain Atlas San Diego, USA2004 [Available from: 770 http://www.brain-map.org/.). Indeed, all details are in the Materials and methods section.

2) In the Materials and methods section, the authors state that exon 1 of Lat2 was replaced with a neomycin cassette. Is there any evidence for an alternative translation start codon downstream of exon 1 that is used in the absence of exon 1. As shown in the UCSC Browser (http://genome.ucsc.edu/cgi-bin/hgTracks?db=mm10&lastVirtModeType=default&lastVirtModeExtraState=&virtModeType=default&virtMode=0&nonVirtPosition=&position=chr5%3A134596883%2D134619265&hgsid=610482073_vc2WSLDW4MAetOvl4BmAD4fijN8N), Lat2 has two different first exons. One exon 1 is entirely noncoding and the other exon 1 includes a 5' UTR and some protein coding sequence. Which exon 1 was deleted? Provide an accession number for your gene structure of Lat2 and a diagram in Figure 1 so that it is clear what was deleted in your mutant mouse.

The reviewer #2 referenced LAT-2 in Chr:5, the wrong gene. LAT2 or SLC7A8 transporter is coded by LAT2 that is a gene located in Chr:14, in both, mouse (Slc7a8) and human (SLC7A8), and neither have alternative translation start codon nor other protein coding in the same locus. For more details please see [http://genome.ucsc.edu/cgi-bin/hgTracks?db=mm10&lastVirtModeType=default&lastVirtModeExtraState=&virtModeType=default&virtMode=0&nonVirtPosition=&position=chr14%3A54722215-54781886&hgsid=647025057_w0nialSkXh3nT2hqaqaI4qoFjLih].

3) Subsection “Localization and quantification of LAT2 in the inner ear”. Provide a reference for "the previous study" and state where the previous study showed localization of LAT2 that you think may be mislocalization. I know you are being courteous here but more information would be helpful. Also considering that there are at least two alternative splice protein coding isoforms of LAT2 (with and without exon 6 of 12), might there be another explanation for the discrepancy? Where was the epitope of LAT2 located in the other study?

As aforementioned, there are not alternative splice protein coding isoforms in the LAT2 (SLC7A8) protein. The most probable explanation is the accuracy of the methodology and equipment used in this manuscript. LAT2 levels are very high in the area surrounding the stria vascularis, what might have led to an erroneous conclusion. Still, the text has been corrected (subsection “Localization and quantification of SLC7A8 in the inner ear”) to: “We observed an intense expression of SLC7A8 in the spiral ligament surrounding the stria indicating that the SLC7A8 epitope (Figure 1—figure supplement 1) is either hidden or absent in the stria vascularis”.

4) Results section, the authors state that there is a significant reduction in latency in the rotorod acceleration test. Latency of what? Similarly (Results section), explain why an "increased exposure to shock on the treadmill" is important and what it means biologically. Just stating it represents "motor coordination performance" and PPI may be useful information for those in the field but to others it is jargon/lingo.

The biological meaning of the behavior test is detailed in Materials and methods section Behavior Test. In addition, in the main results the meaning of the test are described.

Results section: “In contrast, a significant reduction in latency was observed in the rotarod acceleration test indicating impairment in motor coordination in Slc7a8^-/-^ mice (Figure 1—figure supplement 3). Reaffirming poorer motor coordination performance in Slc7a8^-/-^ mice, an increased exposure to shock on the treadmill was also observed (Figure 1—figure supplement 3). Interestingly, a marked impairment was observed in the pre-pulse inhibition of acoustic startle response (PPI), which assesses the response to a high intensity acoustic stimulus (pulse) and its inhibition by a weaker pre-pulse”.

5) Figure 1, panel A, there are no loading controls in the lanes of the Western blot. In the legend it is noted that 50ug of protein was loaded in each lane. That is helpful information but insufficient.

We have included the bactin blots to demonstrate the loading in the western blots of Figure 1. In addition, below we show the original films with anti-SLC7A8 and the stripped membrane with anti- bActin used (Author response image 1 and Author response image 2) to build Figure 1. The red frames indicate the part of the gels represented in Figure 1. Figure 1: On the left side, the western blots of LAT2 are shown, the upper blot in reducing (+DTT) and the bottom one in non-reducing conditions (Ø DTT). On the right side the Western blots of bActin in reducing conditions (up) and non-reducing conditions (bottom) are shown. Author response image 2: On the left side, the western blots of LAT2 in reducing conditions (+DTT) and on the right side the corresponding bActin. Therefore, the lack of expression of LAT2 in the tissues from the *Slc7a8* knockout mice is not due to a deficient loading of the lanes in the gels.

**Author response image 2. respfig2:** 

6) Figure 1, panel B and Results section, the authors need to explain what is "Pre-Pulse inhibition of the acoustic startle response (PPI)" and what is being measured.

See answer in point 4.

7) Figure 1, panels C and F, on the Y-axis, indicate that click thresholds are ABR thresholds.

Data represented are the Auditory Brainstem Response (ABR) threshold in response to click as indicated in the Figure 1 caption. Y-axis of Figure 1 was corrected to: *ABR threshold (dB SPL).*

8) Figure 1, The Y-axis is without a metric/description.

Graph was amended including the metrics of Y axis.

9) Figure 2 also needs a diagram to help orient a reader to the structures and cell types of the inner ear.

The new sketch in the right panel of Figure 2 shows the main structures of the inner ear in the same perspective as the immunofluorescence image of the left panel of Figure 2 for its localization and as an orientation for the readers.

Were the immunofluorescent images acquired and evaluated with the investigator blinded to genotype? Panel B is nice but not sufficient. In panel C, add a key in the figure panel for the three bar types.

Immunofluorescent image capture was carried out using the same laser intensity, time and z-axis thickness for all samples independently of the genotype as described in Materials and methods. In addition, quantification was done using Fiji software macros, a semi-automatized system, being the quantification blinded.

10) Figure 2, panel D, and subsection “Localization and quantification of LAT2 in the inner ear”, which states that there is a progressive increase in LAT2 expression. After P1, the up and down levels of expression seem more like experimental noise and not a progressive increase. How many independent determinations of 3 mice per group of pooled samples were evaluated? Are the error bars for technical replicates?

As described in the Figure 2 caption, mean ± SEM corresponds to a technical triplicate from a pooled sample of 3 independent mouse per group. Reinforcing the results of *RNA* progressive increase, protein expression in young adult and old wild type mice showed the same pattern (see Figure 2, where LAT2 levels were quantified in four cochlear sections from 3 mice per group, thus biological variability corresponds to 3 different mice per group). To clarify the message, the plot showing the progressive increase in *Slc7a8* mRNA expression with age (former Figure 2) is now shown in Figure 2—figure supplement 1.

11) Figure 5, legend, it is not clear what is meant by "[…]unless V60E that[…]" and "[…]is represented the uptake[…]"

Because when V460E mutant was overexpressed in HeLa cells does not reach the plasma membrane. Text was corrected (Figure 5 legend) to: All SLC7A8 variants, except V460E, reached the plasma membrane.

Reviewer #31) Hearing tests in families of patients with LAT2 mutations. It seems prudent to test both the genetics and hearing of family members to determine the genotype-phenotype relationship to ascertain if LAT2 mutations are necessarily pathogenic, and a dosage effect can also be supportive of such a conclusion.

The phenotype we are dealing with is an age-related trait in which inclusion criteria define a threshold of people aged 50 or more (the vast majority of them are in between 60 and 75). This situation implies the following aspects: (1) parents usually are dead, (2) siblings are younger than 50 years-old and thus cannot be included in the study because they won’t be properly classified for the genotype-phenotype relationship leading to a misinterpretation of the results, (3) only few relatives (brothers and sisters) are alive. This is the typical situation of a late onset complex trait in which the absence of a clear pattern of inheritance (i.e. monogenic diseases) and the late onset (i.e. inclusion criteria further limiting the overall number of subjects) making almost impossible to define “families or large familial aggregations”. Nevertheless, being the work carried out on isolated inbred populations, the controls are in some way “related to the patients”. In the same line, the absence of loss-of-function LAT2 mutations in the controls is somehow what the reviewer is asking for.

Also, LAT2 qPCR expression should be assessed in heterozygotes if a dosage effect is proposed.

LAT2 (SLC7A8) protein expression had been evaluated quantifying the levels of the transporter in the cochlea of wild type, heterozygous and knockout mice (see Figure 2). We observed half a dose of the transporter in heterozygous mice.

2) Does histology correlate with hearing loss in mice? The incomplete penetrance and use of C57bl6 to examine an age-related hearing loss candidate gene is problematic. But a correlation between the degree of cell loss and hearing loss can be supportive evidence yet this is lacking. Also for some of the cell markers, assessing the intensity of immunostaining seems odd. One should instead quantify the number of immunostained cells. Lastly, the qPCR results of several genes on the LAT2 were missing.

We address this point in the last answer reviewer #1. There is a good correlation between the hearing loss status and cochlear damage. This is now specified in the text of the manuscript and is shown in (Figure 3—figure supplement 1).